# Durvalumab plus tremelimumab for the treatment of patients with progressive, refractory advanced thyroid carcinoma: the phase II GETNE-DUTHY trial

Single-agent PD-1 blockade demonstrated promising efficacy in advanced thyroid cancer. The phase II, single-arm, multi-cohort GETNE-DUTHY trial (clinicaltrials.gov NCT03753919, EudraCT 2018-001066-42) aimed to determine whether dual anti-PD-L1/CTLA-4 inhibition using durvalumab and tremelimumab can improve the clinical outcomes in advanced thyroid cancer. Three parallel cohorts including patients with differentiated thyroid carcinoma (Cohort 1, $n = 37$), medullary thyroid carcinoma (Cohort 2, $n = 30$), and anaplastic thyroid carcinoma (Cohort 3, $n = 12$) were recruited. Cohort 1 and 2 included patients following progression to previous standard systemic therapy and in Cohort 3 were recruited regardless of previous therapy. The primary endpoint was 6-month progression-free survival rate for Cohort 1 and 2 and 6-month overall survival rate for Cohort 3. Secondary endpoints included objective response rate, progression-free survival, overall survival, and safety. The 6-month progression-free survival rates were 32.4% (95% confidence interval [CI]: 20.4–51.6) (Cohort 1) and 40.8% (95% CI: 26.3–63.6) (Cohort 2); 6-month overall survival rate was 65.6% (95% CI: 43.2–99.8) (Cohort 3). The objective response rates were 8%, 10%, and 33% for Cohort 1, 2, and 3, respectively. No additional safety signals observed. Durvalumab plus tremelimumab treatment in patients with anaplastic thyroid carcinoma met the primary endpoint of this study, showing encouraging survival outcomes.

Thyroid cancer is the most common endocrine malignancy[1] and is characterised by significant histological, clinical, and biological heterogeneity. The histological subtypes of thyroid cancer range from differentiated thyroid carcinoma (DTC), which has a relatively favourable prognosis, to anaplastic thyroid carcinoma (ATC), an aggressive subtype associated with a poor prognosis and a median overall survival (OS) of less than one year[2–4].

Multikinase inhibitors (MKIs) have been established as standard therapies for various histological subtypes of thyroid cancer, with agents such as sorafenib, lenvatinib, and cabozantinib used for the

treatment of DTC[5–7] and vandetanib and cabozantinib used for treating medullary thyroid carcinoma (MTC)[8,9]. However, therapeutic options for patients with DTC who exhibit disease progression on MKIs remain limited[10,11]. Regarding MTC, the recent approval of the selective RET inhibitor selpercatinib for the treatment of RET-mutant tumours and tumour-agnostic NTRK inhibitors (e.g. larotrectinib and entrectinib) for the treatment of non-medullary thyroid cancers with NTRK fusions has transformed the management of advanced disease. However, treatment options for patients who show disease progression while on these agents remain limited[12,13].

✉ e-mail: jcapdevila@vhio.net

Programmed death ligand 1 (PD-L1) inhibitor monotherapy improves the OS of patients with different types of advanced cancers, demonstrating a manageable safety profile in clinical trials[14–17]. However, few studies have evaluated the efficacy of immune checkpoint inhibitors (ICIs), such as anti-PD-1 antibodies, in the treatment of thyroid cancers[18,19]. As the efficacy of ICIs for the treatment of moderately immunogenic tumours is modest, dual targeting of PD-1 and cytotoxic T-lymphocyte antigen 4 (CTLA-4) has emerged as an enhanced therapeutic strategy for other cancers[20,21]. Previous studies on durvalumab (anti-PD-L1) and tremelimumab (anti-CTLA-4) for the treatment of neuroendocrine neoplasms yielded modest but clinically meaningful survival outcomes in a subset of patients with poorly differentiated carcinomas, along with a manageable safety profile[22]. These findings provide a biologically grounded rationale for evaluating the efficacy and safety of dual checkpoint inhibition therapy in patients with advanced thyroid carcinomas, a population with limited therapeutic options and shared molecular vulnerabilities.

In this work, we evaluate the efficacy and safety of dual immune checkpoint inhibition using the combination of durvalumab (anti-PD-L1) and tremelimumab (anti-CTLA-4) in patients with advanced thyroid carcinomas. The study meets its primary endpoint in the cohort of patients with ATC, showing encouraging efficacy in these patients, whereas the clinical benefit is limited in patients with DTC and MTC. Our results support that ICI may play a role in the treatment of patients with the aggressive ATC disease and suggest limited value in patients with DTC and MTC.

## Results

### Patient characteristics

Between April 2019 and October 2023, 88 patients were screened, of whom 79 were included in the study. Of these, 37 had DTC, 30 had MTC, and 12 had ATC (Fig. 1). The MTC cohort was closed prematurely because of slow accrual. All enrolled patients received at least one dose of the study treatment and were included in the full analysis set for efficacy and safety analyses. The baseline demographic and disease characteristics of all patients and the detailed history of systemic therapy for each cohort are summarised in Table 1. In patients with ATC, 33.3% of them were treated with local radiotherapy prior to inclusion. The median patient age was 66 years (range, 32–88 years), and 53.2% of the patients were female. Considering the eligibility criteria, patients in the ATC cohort could be treatment-naïve.

### Efficacy

Outcomes varied across histological subtypes during a median follow-up period of 14.0 months (range, 0.2–61.2). The 6-month PFS rates and median progression-free survival (PFS) were 32.4% (95% confidence interval [CI]: 20.4–51.6) and 3.7 months (95% CI: 2.7–6.5) for patients with DTC; 40.9% (95% CI: 26.3–63.6) and 5.3 months (95% CI: 2.8–23.2) for patients with MTC; and 33.3% (95% CI: 15–74.2) and 3.6 months (95%

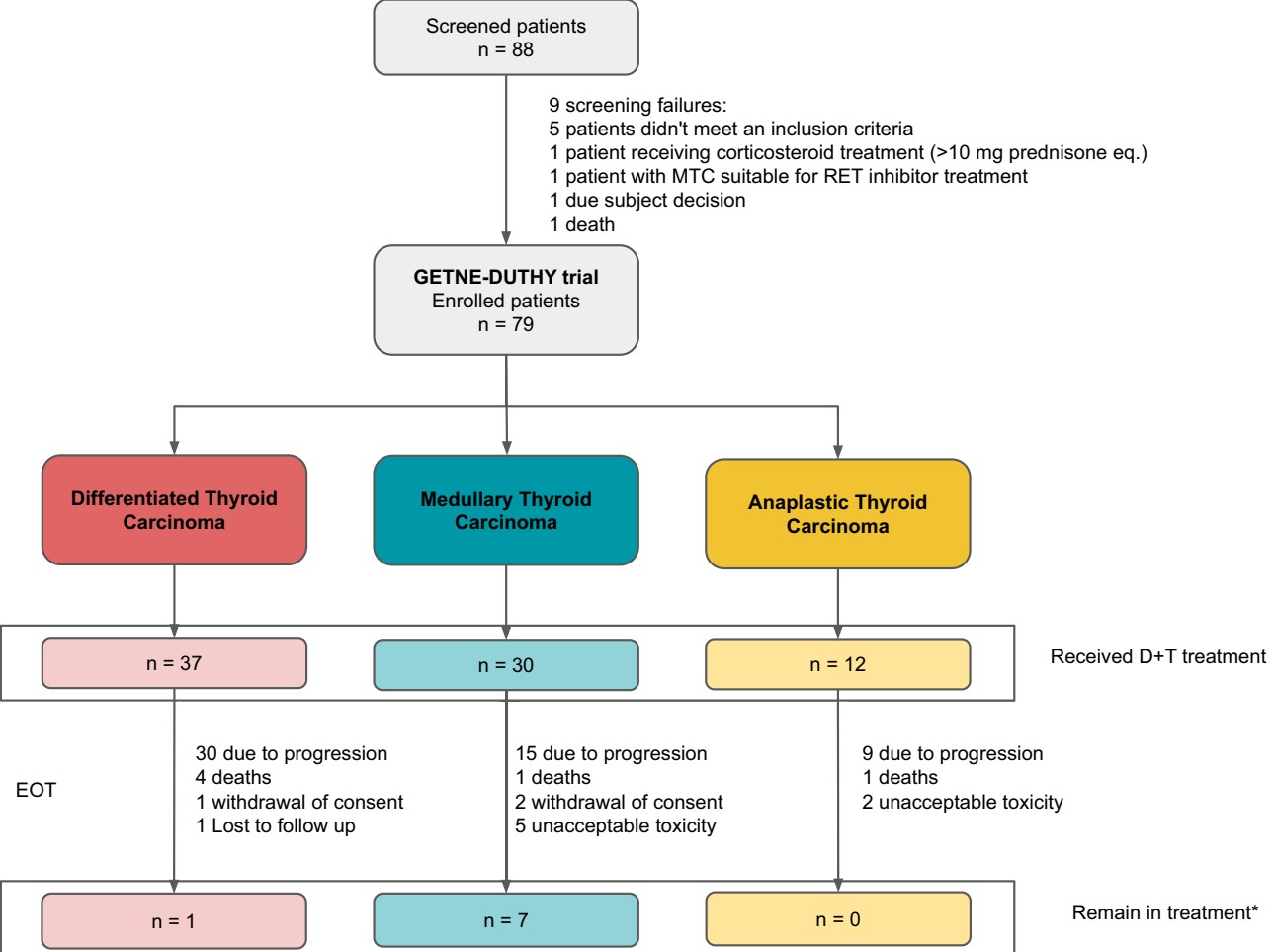

**Fig. 1 | CONSORT diagram of the patient allocation.** The number of patients and reasons for discontinuation of treatment are shown. Patients were allocated in three cohorts of patients with DTC (red), MTC (blue-green) and ATC (yellow). *Patients who continued treatment until the data cutoff. EOT end of treatment, D + T durvalumab plus tremelimumab, MTC medullary thyroid carcinoma.

## Table 1 | Baseline patient characteristics

| Characteristics | DTC n = 37 | MTC n = 30 | ATC n = 12 | GETNE-DUTHY n = 79 |
|---|---|---|---|---|
| Age; median (range) | 67 (48–80) | 58 (32–76) | 71 (54–88) | 66 (32–88) |
| **Sex, n (%)** | | | | |
| Female | 26 (70.3) | 10 (33.3) | 6 (50) | 42 (53.2) |
| Male | 11 (29.7) | 20 (66.7) | 6 (50.0) | 37 (46.8) |
| **ECOG, n (%)** | | | | |
| 0 | 12 (32.4) | 14 (46.7) | 4 (33.3) | 30 (38.0) |
| 1 | 25 (67.6) | 16 (53.3) | 8 (66.7) | 49 (62.0) |
| **Histopathological grade; n (%)** | | | | |
| Well differentiated | 12 (32.4) | 6 (20.0) | 0 (0.0) | 19 (24.1) |
| Poorly differentiated/ Undifferentiated | 10 (27.0) | 1 (3.3) | 11 (91.7)[a] | 22 (27.8) |
| Unknown | 15 (40.5) | 23 (76.7) | 1 (8.3) | 39 (49.4) |
| **Previous surgery[b], n (%)** | | | | |
| Yes | 35[c] (94.6) | 27 (90.0) | 2 (16.7) | 64 (81.0) |
| No | 2 (5.4) | 3 (10) | 10 (83.3) | 15 (19) |
| **Previous radioiodine therapy, n (%)** | | | | |
| Yes | 33 (89.2) | 2 (6.7) | 0 (0.0) | 35 (44.3) |
| No | 4 (10.8) | 28 (93.3) | 12 (100.0) | 44 (55.7) |
| **Previous systemic lines, n (%)** | | | | |
| 0 | 0 (0) | 0 (0) | 7 (58.3) | 7 (10.3) |
| 1 | 12 (32.4) | 20[d] (66.7) | 0 (0.0) | 32 (40.5) |
| 2 | 17 (45.9) | 9 (30.0) | 5 (41.7) | 31 (39.2) |
| ≥3 | 8 (21.6) | 1 (3.3) | 0 (0.0) | 9 (11.4) |
| **Type of previous lines, n (%)** | | | | |
| Chemotherapy | 2 (5.2) | 4 (13.3) | 4 (33.3) | 4 (17.7) |
| Lenvatinib | 34 (91.9) | – | 2 (16.7) | 36 (45.6) |
| Sorafenib | 25 (67.6) | – | – | 25 (31.7) |
| Vandetanib | 2 (5.4) | 30 (100.0) | – | 32 (40.5) |
| Cabozantinib | 1 (2.7) | 7 (23.3) | – | 8 (10.1) |
| RET inhibitors | 1 (2.7) | 3 (10.0) | – | 4 (5.1) |
| Other BRAF/MEK inhibitors | 2 (5.4) | – | 2 (16.7) | 4 (5.1) |
| Other MKIs | 4 (10.8) | – | – | 4 (5.1) |
| **Time from last progression prior study inclusion** | | | | |
| Median; months (95% CI) | 0.8 (0.7–1.1) | 0.9 (0.5–1.3) | 0.8 (NA) | 0.8 (0.7–1.1) |

*ATC* anaplastic thyroid carcinoma, *DTC* differentiated thyroid carcinoma, *ECOG PS* Eastern Cooperative Oncology Group performance status, *MKI* multikinase inhibitor, *MTC* medullary thyroid carcinoma.

[a]All patients with ATC were undifferentiated.
[b]Total or partial thyroidectomy.
[c]In one patient included a laryngectomy and in other consisted of a cordectomy and partial laryngectomy.
[d]Three patients received vandetanib treatment as adjuvant or neoadjuvant therapy.

CI: 2.2–Not Reached [NR]) for patients with ATC (Fig. 2a). The 6-month OS rate and median OS were 65.6% (95% CI: 43.2–99.8) and 13.8 months (95% CI: 5.7–NR), respectively, for patients with ATC (Fig. 2b). The 18-months PFS rates were 6.0% (95% CI: 1.6–22.6), 26.5% (95% CI: 13.6–51.6), and 8.3% (95% CI: 1.3–54.4) for DTC, MTC and ATC, respectively. The PFS and OS outcomes for the three cohorts are presented in Table 2 and Supplementary Tables 1 and 2.

In the DTC cohort, patients with Eastern Cooperative Oncology Group performance status (ECOG PS) of 0 and 1 had median PFS of 8.2 months (95% CI: 2.8–NR) and 2.8 months (95% CI: 2.6–5.5) and median OS of 32.8 months (95% CI: 12.2–NR) and 12.3 months (95% CI: 3.9–28.5), respectively (Supplementary Fig. 1). In the MTC cohort, male

patients had a median PFS of 3.0 months (95% CI: 2.8–15.7) and a median OS of 35.1 months (95% CI: 19.5–NR), whereas female patients had a median PFS of 23.2 months (95% CI: 5.3–NR) and a median OS that was not reached. For the ATC cohort, the median OS was 7.5 months (95% CI: 3.1–NR) for males and 20.8 months (95% CI: 13.8–NR) for females.

The objective response rate (ORR) for the DTC, MTC, and ATC cohorts were 8%, 10%, and 33%, respectively, with no complete response (CR) recorded (Fig. 3 and Supplementary Table 3). The response status at 6 and 12 months were 9% and 8% for the entire cohort, 3% and 3% for the DTC cohort, 10% and 10% for the MTC cohort, and 25% and 17% for the ATC cohort, respectively. The ORR in patients with liver metastasis was 8.3% (1/12). The single patient in this group who showed an objective response had papillary thyroid carcinoma. In the DTC cohort, two patients who showed objective responses had papillary tumours, and one had an oncocytic subtype. Median duration of response (DoR) for the DTC, MTC, and ATC cohorts was 19.3 (interquartile range [IQR] = 18.7), 28.0 (IQR = 13.2), and 5.7 (IQR = 14.9) months, respectively.

Nine patients survived for at least four years from treatment initiation (Supplementary Fig. 2), and eight were still alive at the time of data cutoff (May 2024). Of these nine patients, three had DTC. All three patients with DTC had liver metastases at the time of study enrolment and a history of radioiodine therapy. Among the long-term survivors with DTC, one patient with follicular histology, who was initially diagnosed with metastatic disease, showed disease progression and was ongoing, the second patient with follicular histology showed disease progression and subsequently died, and the patient with oncocytic histology remained progression-free throughout the study observation period. Four long-term survivors had MTC. Of these, one patient remained progression-free and was ongoing, whereas the other three showed disease progression and were alive at the time of the data cutoff. The two long-term survivors with ATC showed disease progression but were still alive, with survival times of 55.4 and 57.2 months, respectively. Among the long-term survivors who showed disease progression, one patient with ATC did not receive subsequent treatment, whereas the other seven received different treatments, mostly MKIs. The details of all treatment regimens administered to the patients after progression to the study treatment are shown in Supplementary Table 4.

No significant difference in 6-month PFS was observed between patients with and without liver metastasis (34.8% [95% CI, 19.9–80.9] for patients with liver metastasis vs. 39.5% [95% CI, 28.0–55.9] for those without liver metastasis). Similarly, the 6-month OS rates did not differ significantly between the two groups (75.6% [95% CI, 60.3–94.7] vs. 81.1% [95% CI, 71.1–92.4] for patients with and without liver metastasis, respectively) (Supplementary Fig. 3). Although patients with immune-mediated adverse events (imAEs) had numerically lower hazard ratios (HR) for both PFS and OS, these associations did not reach statistical significance when analysed as time-dependent variables (PFS: HR = 0.75, 95% CI: 0.45–1.25, p = 0.273; OS: HR = 0.61, 95% CI: 0.34–1.10, p = 0.103).

Among the 10 patients who showed treatment responses, four (40%) were PD-L1 positive (Fig. 3, Supplementary Data 1). Of the ten patients, one with oncocytic carcinoma had microsatellite instability (MSI) and was PD-L1 negative. Among patients with ATC who showed treatment responses, two (50%) were PD-L1 positive and two did not have a sufficient amount of tumour samples to allow for the assessment of PD-L1 status. RET status was locally assessed in 15 patients with MTC (50.0%), of whom 11 (73.3%) had mutations. Data on RET status were available for one patient with MTC who achieved an objective response, and the data showed no RET mutations. In the ATC cohort, data on BRAF status were available for eight (66.7%) patients. Notably, among the four patients who achieved an objective response, only two had available data on BRAF status: one with *BRAF* wild type and the

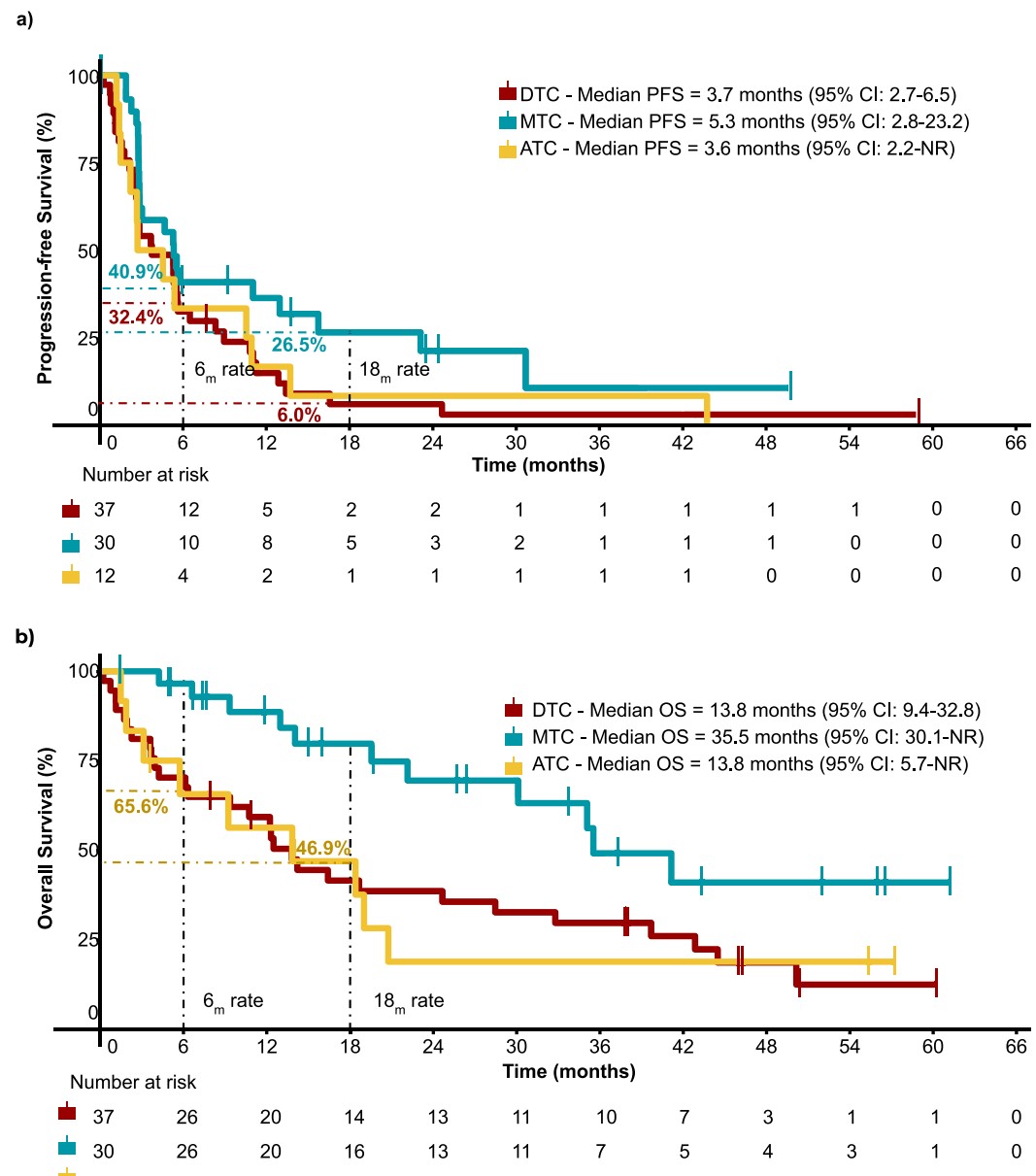

**Fig. 2 | Kaplan–Meier curves of the overall population stratified by cohort.**
**a** Progression-free survival (PFS) of patients with DTC, MTC, and ATC. The coloured dashed lines represent the 6-month PFS and 18-month PFS rates for patients with DTC (red) and MTC (blue-green). **b** Overall survival (OS). The yellow dashed line represents the 6-month OS and 18-month OS rates for patients with ATC. Source data is provided as a Source data file. $6_m$ rate 6-month rate, $18_m$ rate 18-month rate, HR hazard ratio.

other with *BRAF^{V600E}*. BRAK/MEK inhibitors were administered prior to two patients with ATC, one for a duration of one week and another for 11 months. Durvalumab and tremelimumab provided in these two patients a PFS interval of 3 and 10 months after BRAF/MEK inhibitors, respectively.

## Safety profile
At the time of the data cutoff, eight patients (10.1%) remained on study treatment. The reasons for treatment discontinuation included disease progression (54 patients [68.4%]), unacceptable toxicity (seven patients [8.9%]), death (six patients [7.6%]), patient's decision (three patients [3.8%], two of them withdraw the consent), and loss to follow-up (one patient [1.3%]). The median number of treatment cycles administered to all treated patients was five (range: 1–66) for durvalumab and four (range: 1–4) for tremelimumab. Regarding the management of treatment-related adverse events (TRAEs), durvalumab

treatment was delayed in 12 patients (15.2%), whereas tremelimumab treatment was delayed in seven patients (8.7%). Durvalumab cycles were omitted in four patients (5.1%), whereas tremelimumab cycles were omitted in two patients (2.5%).

A total of 49 patients died. Of these, 39 (79.6%) died due to disease progression and seven (14.3%) due to adverse events unrelated to treatment. The cause of death was unspecified in two (4.1%) patients. One patient died due to dyspnoea; notably, a causal relationship with the study treatment could not be established. The most frequent TRAEs observed in the entire study cohort were dermatological events (44.3% of patients), fatigue (20.3%), diarrhoea (15.2%), and elevated hepatic enzyme levels (11.4%) (Fig. 4 and Supplementary Table 5). All TRAEs were managed with standard supportive measures. The toxicity profiles of the treatments were comparable across cohorts, and no additional safety signals were observed. Grade ≥3 TRAEs occurred in 19.0% of patients, with the most frequent being fatigue (2.5%),

**Table 2 | Progression-free survival (PFS), overall survival (OS), and duration of response (DoR) for DTC, MTC, and ATC cohorts**

| Outcomes | DTC n = 37 | MTC n = 30 | ATC n = 12 |
|---|---|---|---|
| Median PFS; months (95% CI) | 3.7 (2.7–6.5) | 5.3 (2.8–23.2) | 3.6 (2.2–NR) |
| 6-month PFS rate; % (95% CI) | 32.4 (20.4–51.6) | 40.9 (26.3–63.6) | 33.3 (15.0–74.2) |
| 18-month PFS rate; % (95% CI) | 6.0 (1.6–22.6) | 26.5 (13.6–51.6) | 8.3 (1.3–54.4) |
| Median OS; months (95% CI) | 13.8 (9.4–32.8) | 35.5 (30.1–NR) | 13.8 (5.7–NR) |
| 6-month OS rate; % (95% CI) | 70.3 (56.7–86.7) | 96.6 (90.1–100) | 65.6 (43.2–99.8) |
| 18-month OS rate; % (95% CI) | 41.5 (28.0–61.4) | 79.8 (65.3–97.5) | 46.9 (25.0–87.9) |
| Median DoR; months (IQR) | 19.3 (18.7) | 28.0 (13.2) | 5.7 (14.9) |

NR not reached.

encephalitis (2.5%), and arthritis (2.5%). Serious adverse events (SAEs) were experienced by 43 (54.4%) patients, and they mainly included dyspnoea (6.3%), diarrhoea (6.3%), fever (3.8%), and fatigue (3.8%). Grades 1–2 SAEs were experienced by 14 patients (17.7%), 50% of whom may have been related to COVID-19. Grade ≥3 imAEs occurred in six (7.6%) patients (Supplementary Table 6).

## Discussion

The phase II GETNE-DUTHY trial is the first prospective study to evaluate the efficacy of durvalumab plus tremelimumab in treating major histological subtypes of advanced thyroid carcinoma. The study met its primary endpoint in the ATC cohort, demonstrating a 6-month OS rate of 65.6%. Considering the known poor prognosis of ATC and the limited efficacy of currently available systemic therapies, the outcomes observed in the ATC cohort are particularly encouraging.

Although chemotherapy regimens and targeted therapies, including BRAF/MEK inhibitors for patients with BRAF[V600E] mutations, are utilised in frontline settings, their outcomes remain dismal for most patients[23]. The results of the present study, which indicated a median OS of 13.8 months and an 18-month survival rate of 46.9%, highlight a clinically meaningful beneficial effect of ICIs on overall survival in patients with ATC, regardless of whether they have BRAF[V600E] mutations. The two patients with ATC who received prior BRAF/MEK inhibitors showed promising PFS with durvalumab and tremelimumab, suggesting ICI is yet a valuable option after failure of MKIs in these patients. Recent phase II trials have demonstrated the clinical efficacy of ICI in ATC. Spartalizumab, an anti-PD-1 agent, achieved an ORR of 19% in a phase II clinical trial[19], whereas the combination of nivolumab (anti-PD-1) and ipilimumab (anti-CTLA-4) achieved an ORR of 30% in an exploratory cohort of patients with ATC[24], which is comparable to the ORR observed in the present study. However, cross-trial comparisons of the outcomes of these therapies should be interpreted cautiously. The ORR observed in the present study was numerically higher than that reported for single-agent PD-1 blockade therapy for ATC and comparable to that of another dual ICI regimen. These findings suggest that dual checkpoint inhibition may offer clinical benefits to patients with ATC. In addition to ICI monotherapy and dual blockade, combinations of targeted therapies have shown promise in ATC treatment. The combination of an anti-PD-1 agent with BRAF/MEK inhibitors or other MKIs yields objective responses in patients with ATC, often with a manageable safety profile[25,26]. For instance, in a small, single-centre cohort study, the addition of ICI to BRAF/MEK inhibitors yielded a substantial benefit in patients with BRAF[V600E] mutations, with the patients achieving a median OS of 43 months[27]. In other studies, the combinations of cabozantinib plus atezolizumab or lenvatinib plus pembrolizumab achieved an ORR of 21.4% and 66%, respectively[26,28], with the median OS for patients treated with lenvatinib plus pembrolizumab being 18.5 months. The combination of immunotherapy and MKIs for ATC treatment is currently being investigated. Moreover, ongoing studies are investigating the efficacy of lenvatinib plus nivolumab

(NCT05696548) and surufatinib plus tislelizumab (NCT04579757). Our results add to the growing evidence that ICIs, alone or in combination with other drugs such as MKIs, constitute a therapeutic strategy that may alter the clinical trajectory of ATC management.

The efficacy of the study treatment in the DTC and MTC cohorts was modest, with 6-month PFS rates of the groups being 38.8% and 35.1%, respectively. In the DTC cohort, the efficacy of durvalumab plus tremelimumab was within the range of outcomes reported for other dual ICI combinations and did not appear to improve the outcomes of single-agent ICI therapy[18,24]. These findings underscore the fundamental limitations of current ICI combination strategies. Thus, with the available data, a benefit of adding anti-CTLA-4 to anti-PD-1 treatment cannot be confirmed or discarded. Moreover, re-evaluation of immunotherapy as a single therapeutic strategy for these carcinomas may be warranted. MKIs, which have demonstrated efficacy and remain the cornerstone of treatment for radioiodine-refractory advanced DTC, are potential options for combined treatment strategies[6,29]. However, existing evidence suggests that combining MKIs with ICI does not significantly enhance clinical outcomes. For instance, lenvatinib combined with pembrolizumab does not improve ORR and PFS compared to lenvatinib alone[6,30]. Prospective studies are needed to determine whether sequential or concurrent ICI–MKI regimens can overcome acquired resistance following failure of MKI therapy and to ascertain whether biomarker-driven patient selection should be prioritised.

In this study, durvalumab plus tremelimumab showed modest efficacy in the MTC cohort, with survival outcomes inferior to those previously reported for MKIs[31–34]. In a previous exploratory study, a cohort of patients with MTC who received nivolumab plus ipilimumab showed no treatment response, indicating the limited benefit of dual ICI therapy for MTC[24]. The limited availability of studies evaluating treatment options for patients who show disease progression while on standard therapies constrains the interpretation of our findings. Therefore, further studies are required to determine the potential role of immunotherapy in MTC management.

Selpercatinib, a RET inhibitor that achieved an ORR of 69.4% (HR = 0.28; 95% CI, 0.16–0.48) in a recent study, has been added to the therapeutic repertoire for thyroid cancer[35]. This development substantially changed the standard of care for patients with cancer since the conceptualisation of the GETNE-DUTHY trial. In light of these changes, enrolment was discontinued after the inclusion of 30 of the initially planned 36 patients. However, the cohort was sufficiently large to provide meaningful insights into the treatment of patients with MTC. In this context, defining the efficacy of subsequent therapeutic options after disease progression while on RET inhibitors represents a critical unmet need, particularly given the absence of prospective evidence supporting the sequential use of cabozantinib or vandetanib in this clinical scenario. Notably, of the four patients in the present study who were previously treated with RET inhibitors, none showed objective responses. This finding suggests the limited efficacy of ICIs in this setting; however, the numbers supporting this result were too small to draw definitive conclusions.

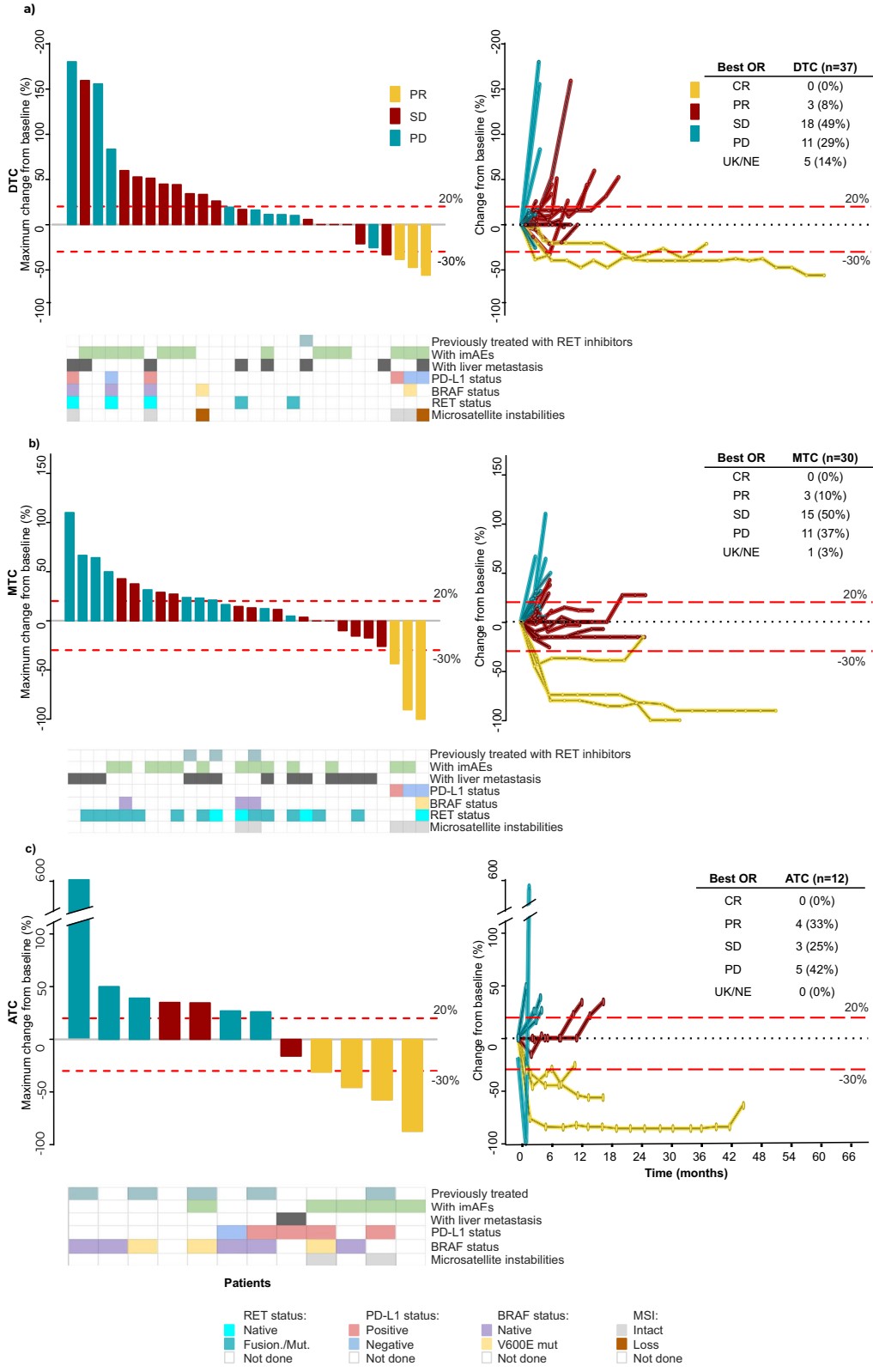

**Fig. 3 | Waterfall and spider plots summarizing the responses achieved with durvalumab plus tremelimumab.** Patients with **a** differentiated thyroid carcinoma (DTC), **b** medullary thyroid carcinoma (MTC), and **c** anaplastic thyroid carcinoma (ATC). Representation of the best response for each patient in a waterfall plot (left) and a spider plot showing the change in baseline tumour size from the first to last tumour evaluation for each patient (right), stratified according to the study cohorts. Responses were classified according to RECIST v1.1. Patients without target lesions, without measurable diameters, or with a "UK" or "NE" best overall response (OR) were excluded from waterfall plots. The results of the best OR were reported according to the number and percentage of patients with respect to the total number of patients in the cohort. Source data is provided as a Source data file. PR partial response (yellow), SD stable disease (red), PD progression disease (blue-green), UK/NE unknown/not evaluable, imAE immune-mediated adverse event, MSI microsatellite instability.

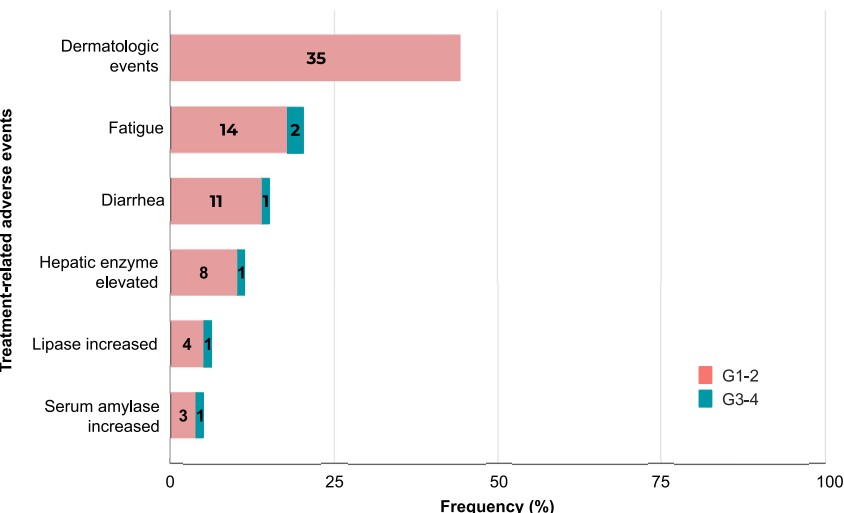

**Fig. 4 | Toxicity profile of durvalumab plus tremelimumab in patients with advanced thyroid carcinomas.** Most frequent treatment-related adverse events (TRAEs) with an overall threshold of 5% and the number of patients who experienced TRAEs, stratified by grade 1 or 2 (red) and grade 3 or 4 (blue-green). Source data is provided as a Source data file.

In other malignancies, liver metastases are associated with poor outcomes following ICI therapy[36]. However, in our cohort of patients with thyroid cancer, no statistically significant association between liver metastases and patient outcomes was found. Notably, exploratory biomarker analyses were performed with a subset of patients to identify potential predictors of the response. These exploratory preliminary results suggest that emerging biomarkers, including PD-L1 expression and MSI status, may help refine patient selection, although further research will be needed. Early data from a trial on pembrolizumab therapy for DTC suggested an enhanced clinical benefit in patients with PD-L1-positive tumours[37]; however, this association between PD-L1 expression and patient outcomes was not confirmed in subsequent phase II trials[19].

Consistent with previous reports, no statistically significant association between the occurrence of imAEs and PFS was observed in the present study[38]. This suggests that the occurrence of imAEs may not directly predict PFS in patients with thyroid cancer. However, optimising the management of imAEs, particularly mild events, is crucial for minimising treatment interruptions and maintaining patients on therapy, which may contribute to improved overall outcomes of patients.

The toxicity profile observed in the present study was manageable and consistent with those in previous reports[22,39]. The most common adverse events observed in this study were dermatological events, fatigue, diarrhoea, and elevated hepatic enzyme levels, which are consistent with the safety profiles previously reported for ICI regimens for thyroid cancer and other solid tumours[18,19,24]. Treatment discontinuation due to unacceptable toxicity occurred in 8.9% of the patients in this trial, which is comparable to the 9.7% reported for patients with thyroid carcinoma treated with pembrolizumab[18].

This study had several limitations. First, PD-L1 status was not systematically assessed in all patients; thus, we could not reach definitive conclusions regarding its predictive value. Future studies are needed to comprehensively evaluate the molecular biomarkers of response to ICIs in patients with thyroid cancer, particularly those with ATC. Second, we inherently relied on historical outcome data for comparisons and sample size assumptions, particularly for the DTC and MTC cohorts treated in later-line settings, where published data on outcomes beyond second-line therapy are limited. This may have led to an overestimation of the expected efficacy rates compared with those of a prospectively randomised control arm. At the time of the study design, only the COSMIC-311 trial provided data on benchmark PFS rates for placebo-treated populations who showed progression after treatment with lenvatinib or sorafenib, indicating a benchmark PFS that supported our initial assumptions. Third, most patients with DTC or MTC in this study were heavily pretreated (50.6% underwent ≥2 previous systemic therapies). Additionally, the scarcity of post-second-line therapy data complicates the interpretation of these findings. Fourth, patients who showed treatment response were included in the biomarker analyses; therefore, the possibility of post-hoc selection bias cannot be completely excluded. However, these analyses were exploratory, and the results were interpreted cautiously.

This phase II trial, in which Simon's two-stage design was employed to determine the sizes of the DTC and MTC cohorts, provides valuable prospective data and insights into the methodological challenges of using a clinical trial design to evaluate progressive thyroid carcinomas. Although Simon's design remains a valid framework for efficiently identifying agents with high efficacy, its stringent statistical thresholds may limit the evaluation of therapies that demonstrate moderate yet potentially clinically meaningful efficacy in patients with limited treatment options. Based on the results of this trial, particularly regarding therapies for patients with limited therapeutic options, such as those with advanced thyroid cancers, we suggest that alternative study designs that balance hypothesis testing and therapeutic discovery may be more appropriate for future studies. Exploratory single-arm designs with pre-specified criteria for identifying signals of activity that do not rely solely on stringent historical comparisons or innovative Bayesian adaptive methodologies should be prioritised to avoid prematurely abandoning the evaluation of therapeutic agents with promising, albeit potentially moderate, clinical efficacy.

In conclusion, the GETNE-DUTHY trial demonstrated that durvalumab plus tremelimumab shows encouraging efficacy in the treatment of patients with ATC, highlighting the potential role of ICIs in the treatment of this aggressive disease. The safety profile of the combination therapy was manageable and consistent with the established ICI toxicity expectations. Although this study provides valuable prospective data that can inform the design of future trials on the treatment of patients with DTC and MTC, the modest efficacy results observed in these cohorts suggest that further development of the ICI combination used in this study may have limited value. Therefore, future studies are needed to explore alternative or combination therapeutic strategies for these thyroid cancer subtypes.

## Methods

### Trial design and patients

This trial was conducted in accordance with the principles of the Declaration of Helsinki and the International Conference on Harmonization Guidelines for Good Clinical Practice. The study protocol was approved by the competent authority in Spain (Agencia Española de Medicamentos y Productos Sanitarios, AEMPS) and the independent ethics committee of Vall d'Hebron University Hospital on 20 February 2019 (Ref. 2019/369). Written informed consent was obtained from all patients prior to their enrolment in the study.

The DUTHY/GETNE T-1812 trial (EudraCT: 2018-001066-42; ClinicalTrials.gov: NCT03753919; published 22 November 2018) was a phase II, single-arm, multi-cohort trial with three parallel cohorts conducted by the Grupo Español de Tumores Neuroendocrinos y Endocrinos (GETNE) in 15 reference centres in Spain. Patients with confirmed locally advanced or metastatic thyroid carcinoma were recruited into the following three cohorts: patients with DTC (including papillary, follicular, poorly differentiated, and oncocytic tumours) who showed disease progression to systemic therapy with lenvatinib or at least two MKIs, including or not lenvatinib (Cohort 1); patients with MTC who showed disease progression to systemic therapy with vandetanib or at least two MKIs, including or not vandetanib (Cohort 2); and patients with ATC regardless of prior therapy (Cohort 3). Data on the use of cabozantinib for DTC and selpercatinib for MTC were not available at the time of study design. The primary inclusion criteria for this study were as follows: age ≥18 years, Eastern Cooperative Oncology Group (ECOG) performance status of 0 or 1, body weight >30 kg, life expectancy >3 months, and adequate organ and bone marrow function. Female patients were required to be postmenopausal or have a negative urinary or serum pregnancy test within 72 h prior to the first dose of the study treatment. The main exclusion criteria were previous treatment with a PD-1, PD-L1, or CTLA-4 inhibitor; history of immunotherapy; immunodeficiency or use of any immunosuppressive medication within 28 days before the first dose of durvalumab/tremelimumab, with the exception of intranasal and inhaled corticosteroids or systemic corticosteroids at physiological doses not higher than 10 mg/day of prednisone or its equivalent; concurrent use of cancer treatments or investigational products; and a history of allogeneic organ transplantation. A complete list of eligibility criteria is provided in the study protocol.

### Procedures

Patients received durvalumab 1500 mg intravenously every four weeks plus tremelimumab 75 mg intravenously every four weeks for up to four cycles, followed by durvalumab monotherapy every four weeks until disease progression was confirmed according to Response Evaluation Criteria In Solid Tumours (RECIST) v1.1, unacceptable toxicity, or withdrawal of consent, whichever occurred first (Supplementary Fig. 4). The dosages were selected using pharmacokinetic models developed with data from a phase 1 study for durvalumab[40], and phase 1 to 3 for tremelimumab[41]. The rationale for the selected scheme was provided by previous clinical trials with dual ICI in patients with advanced melanoma and phase 1 trials with durvalumab plus tremelimumab[42]. Dose delays (≤12 weeks) or omissions were permitted for the management of toxicity. Dose reduction was not permitted.

Clinical assessments, including a complete physical examination, measurement of vital signs, laboratory tests (blood chemistry, haematology, and thyroid function), and evaluation of ECOG performance status, were performed during the initial screening, every two weeks during treatment with durvalumab plus tremelimumab, and every four weeks during maintenance with durvalumab monotherapy until the end of treatment. All patients provided archival formalin-fixed paraffin-embedded tumour tissue samples. Blood samples for biomarker analysis were collected during screening visits, on day one of the second treatment cycle, and at the end of treatment. Tumour response was assessed using computed tomography or magnetic resonance imaging scans and categorised according to RECIST 1.1. Tumour response was assessed at baseline and every 12 weeks until disease progression or death was confirmed, irrespective of dose delays.

### Outcome measures

The primary study endpoint was the 6-month progression-free survival (PFS) rate for the DTC and MTC cohorts and the 6-month OS rate for the ATC cohort. For the DTC and MTC cohorts, the primary endpoint was considered met if a 6-months PFS rate of 45% was achieved. For the ATC cohort, the primary endpoint was considered met if a 6-months OS of 35% was achieved. The 6-month PFS rate was defined as the percentage of patients who remained alive and disease progression-free at 6 months, whereas the 6-month OS rate was defined as the percentage of patients who remained alive at six months after the start of the study treatment.

The secondary endpoints were objective response rate (ORR), defined as patients who achieved partial response (PR) or complete response (CR); duration of response (DoR), defined as the time from the first PR/CR to disease progression or the last tumour assessment; median PFS, defined as the time from the start of treatment until disease progression or death (patients without events were censored at the last tumour assessment); median OS, defined as the time from the start of treatment until death (patients without events were censored at the last follow-up); response status at 6 and 12 months after the start of treatment, defined as the percentage of patients who showed treatment response at each timepoint; 18-month PFS rate, defined as the percentage of patients who remained alive and disease progression-free at 18 months; 18-month OS rate, defined as the percentage of patients who remained alive at 18 months after the start of the study treatment; and safety profile. Tumour response was assessed using the RECIST 1.1 and irRECIST 1.1. No significant differences were observed between the two evaluation methods (Supplementary Table 3); therefore, the results reported in this study were evaluated using the RECIST 1.1. The prognostic value of biomarkers was established as an exploratory objective. As liver metastasis has been related to poorer outcomes in patients with solid cancers treated with immunotherapy[36], a post-hoc analysis of the prognostic value of the presence of liver metastases was included.

For safety analysis, all treatment-related adverse events (TRAEs) were coded and graded according to the National Cancer Institute's Common Terminology Criteria for Adverse Events (NCI-CTCAE), version 5.0.

### Molecular assessments

Different exploratory analyses of biomarkers were performed to further categorise patients who responded to treatment. PD-L1 staining was performed using the VENTANA PD-L1 (SP263) assay (Cat. No. 741-4905/07419821001; Ventana Medical Systems, Inc. Roche) on the VENTANA BenchMark ULTRA automated staining platform, using the OptiView DAB IHC detection kit (Cat. No. 760-700/06396500001). All immunohistochemically stained sections were reviewed and scored by a board-certified pathologist. PD-L1 expression was assessed using the combined positive score (CPS), with positivity defined as CPS ≥ 1. CPS was calculated as the number of PD-L1-positive tumour and immune cells divided by the total number of viable tumour cells, multiplied by 100. Central revision of the microsatellite instability (MSI) phenotype using immunohistochemistry (IHC) was performed for patients who showed an objective response according to the following ESMO recommendations: loss of nuclear expression of at least one of the following proteins: MLH1 (Roche, cat n°: 760-5091), MSH6 (Roche, cat n°: 760-5092), PMS2 (Roche, cat n°: 760-5094), and MSH2 (Roche, cat n°: 760-5093). BRAF$^{V600}$ was locally assessed at each site using next-generation sequencing (NGS), IHC, real-time PCR, or the Idylla BRAF Mutation Test. The RET M918T mutation in patients with MTC and RET fusions in patients with DTC were assessed locally using NGS.

## Statistical analysis

Simon's two-stage design (Minimax) was used for sample size estimation in this study[43]. Considering the information in previous studies[6], the null hypothesis assumed a 6-month PFS rate of 25% for the DTC and MTC cohorts based on historical data, whereas the target for the experimental therapy was a 45% rate. Calculations using a one-sided α of 5% and 80% power showed that 36 patients per cohort were required. Per Simon's criteria, progression to the second stage required ≥5 of the first 17 patients per cohort to be progression-free at 6 months. A single-stage design was implemented for the ATC cohort. Assuming a historical 6-month OS rate of 5% and an anticipated improvement to 35% (one-sided α = 5%, power = 80%), 12 patients were needed and ≥3 patients alive at 6 months to declare the study positive.

Efficacy and safety analyses were conducted using the full analysis set, which included all patients. The response rates were calculated as the percentage of patients who achieved a response and their 95% CI was estimated using the Clopper–Pearson method. PFS and OS were estimated using the Kaplan–Meier method and Cox proportional model, and comparisons were tested using log-rank tests (Mantel-Cox). The 18-month PFS rate was defined as the percentage of patients who remained alive and disease progression-free at 18 months, whereas the 18-month OS rate was defined as the percentage of patients who remained alive at 18 months after the start of the study treatment. Patients without event at the data cutoff point were censored at the last tumour assessment for PFS and at the last follow-up for OS analysis. Although the protocol planned the estimation of primary endpoints using binomial calculations, some tumour assessments and survival follow ups were performed within +/− 1 month from the 6 month expected timepoint due to changes in scheduled visits per patient or site availability, or different clinical courses of the disease, so the sponsor decided that Kaplan–Meier estimated would give a more accurate estimation at 6 months. Binomial calculations are listed in Supplementary Table 7. The presence of immune-mediated adverse events (imAEs) was introduced as a time-dependent variable in a Cox proportional model to analyse the relationship between imAEs and PFS. Primary statistical tests were one-tailed and other statistical tests used for efficacy and safety analyses were two-tailed, and results with $p$-values < 0.05 were considered significant. Data were collected through the MFAR eCRF system. All statistical analyses were performed using R (version 3.6.3 [2020-02-29]; The R Foundation for Statistical Computing, Vienna, Austria).

## Reporting summary

Further information on research design is available in the Nature Portfolio Reporting Summary linked to this article.

# Data availability

The data that support the findings of this study are under restricted access for equivalent purposes for which the patients granted their consent to use the data (i.e. for research in thyroid carcinomas). Access can be obtained from the corresponding author (jcapdevila@vhio.net) upon request. Data will be provided anonymously, with no personal identifiable data. Data are available within the Article and Supplementary Information. Source data are provided with this paper.

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

## Acknowledgements

The authors thank all patients and families, investigators and study staff involved in the GETNE-DUTHY study; the MFAR Clinical Research team for regulatory, monitoring, and quality assurance activities; Guillem Marco Ph.D. and Pau Doñate Ph.D. for manuscript and language editing; and Emilio Pecharroman M.Sc. for statistical support. This study was sponsored by the Grupo Español de Tumores Neuroendocrinos y Endocrinos (GETNE). AstraZeneca awarded a grant to GETNE to cover the study costs and provided the study medication. The funder had no role in the design or conduct of the study.

## Author contributions

J.C. conceived and designed the trial. All authors contributed in formal analysis, funding acquisition, investigation, project administration, resources, supervision, validation, visualization, and writing and finally approving of this paper.

## Competing interests

The authors declare the following competing interests: A.G.A. declares speakers' bureau from EISAI Europe and Lilly; and travel, accommodations and expenses from Advanz, EISAI, Ipsen, ADACAP (Novartis), Amgen, Pfizer, Lilly. M.P.S. declares honoraria and payment for travel/accommodations/expenses from Eisai Europe, Merck Serono, and MSD. C.A.E. declares honoraria as speaker fees, served on advisory boards, and received sponsorship and accommodation at scientific meetings from Astra Zeneca, IPSEN, Pfizer, Bayer, and MSD. T.A.G. declares fees for speakers, consultancy, research, and other nonfinancial support from IPSEN, Eli Lilly, Adacap, Pfizer, EISAI, Bayer, Johnson & Johnson, Astellas-Pharma, Roche, BMS, and MSD. A.C.B. declares research and speaking grants from Esteve, speaking grants from Lilly, and travel grants from MSD, Roche and Amgen. N.B. declares consulting honoraria from Merck Serono, Bristol-Myers Squibb, Lilly, and Eisai; and travel/accommodation expenses from Merck Serono. E.S.G. declares honoraria as a consultant for GSK and research grants (institution) from GSK and Rgenta Therapeutics. The remaining co-authors declare no competing interests regarding this study.

## Additional information

Jaume Capdevila [1,2] ✉, Jorge Hernando[1,2], Javier Molina-Cerillo[3], Maria Plana Serrahima [4], Miren Taberna Sanz[4], Beatriz Castelo[5], Cristina Álvarez-Escolá[6], Alberto Carmona-Bayonas [7], Inmaculada Ballester Navarro[7], Lara Iglesias[8], Mateo Bover [8], Alejandro Garcia-Alvarez [1,2], Javier Lavernia[9], Ricardo Yayá Tur[9], Sofia Ruiz[10], Neus Baste[11], Isabel Lorenzo-Lorenzo[12], Alfredo Sanchez-Hernandez [13], Enrique Sanz-Garcia [14,15], Gloria Marquina [16], Paolo Nuciforo [17], Belén Elguero[2], Enrique Grande[18] & Teresa Alonso-Gordoa[3]

[1]Department of Medical Oncology, Gastrointestinal and Endocrine Tumor Unit, Vall d'Hebron Hospital Universitari, Vall d'Hebron Barcelona Hospital Campus, Barcelona, Spain. [2]Neuroendocrine and Endocrine Tumor Translational Research Program (NET-VHIO), Vall d'Hebron Institute of Oncology (VHIO), Vall d'Hebron Barcelona Hospital Campus, Barcelona, Spain. [3]Medical Oncology, Hospital Universitario Ramón y Cajal, Madrid, Spain. [4]Medical Oncology Department, Catalan Institute of Oncology (ICO), IDIBELL, ONCOBELL, L'Hospitalet de Llobregat, Barcelona, Spain. [5]Medical Oncology Department, University Hospital La Paz, Madrid, Spain. [6]Endocrinology Department, University Hospital La Paz, Madrid, Spain. [7]Medical Oncology Department, Hospital Universitario Morales Meseguer, Universidad de Murcia (UMU), IMIB, Murcia, Spain. [8]Medical Oncology Department, Hospital Universitario 12 de Octubre, Madrid, Spain. [9]Medical Oncology Department, Hospital Fundación IVO (Instituto Valenciano de Oncología), Valencia, Spain. [10]UGCI of Medical Oncology, Hospitales Regional y Universitario Virgen de la Victoria, IBIMA, UMA, Málaga, Spain. [11]Medical Oncology Department, Hospital Clínic de Barcelona, IDIBAPS, Barcelona, Spain. [12]Medical Oncology Service, Complejo Universitario Hospitalario de Vigo (CHUVI) Alvaro Cunqueiro, Vigo, Pontevedra, Spain. [13]Medical Oncology Service, Hospital Provincial de Castellón, Castellón de La Plana, Spain. [14]Medical Oncology Department, Centro Integral Oncológico "Clara Campal", Hospital Universitario HM Sanchinarro, Madrid, Spain. [15]Division of Medical Oncology and Hematology, Princess Margaret Cancer Centre, Toronto, Canada. [16]Medical Oncology Department, Hospital Clínico Universitario San Carlos, Department of Medicine, School of Medicine, Universidad Complutense de Madrid (UCM), IdISSC, Madrid, Spain. [17]Molecular Oncology Group, Vall Hebron Institute of Oncology (VHIO), Vall d'Hebron Barcelona Hospital Campus, Barcelona, Spain. [18]Medical Oncology Department, MD Anderson Cancer Center Madrid, Madrid, Spain. ✉e-mail: jcapdevila@vhio.net

