## [Transparent Peer Review file · Nature Communications]

Durvalumab plus tremelimumab for the treatment of patients with progressive, refractory advanced thyroid carcinoma: The phase II GETNE-DUTHY trial.

Corresponding Author: Dr Jaume Capdevila

Version 0:

Reviewer comments:

Reviewer #1

(Remarks to the Author)

This is a very interesting phase II, single-arm, multi-cohort trial was conducted to 70 determine whether dual anti-PD-L1/CTLA-4 inhibition using durvalumab and tremelimumab can improve the clinical outcomes in advanced thyroid cancer. The results of this trial are specifically meaningful to the ATC cohort of patients particularly the BRAF wild type ones for who no meaningful drugs are available.

I have provided specific questions in the attached edited version of manuscript but the main points to focus on are:

1. The methods state that 18 month PFS would be reported for ATC but the results is missing that. Honestly, having methods above results is preferred mainly because one can refer results to methods and make sure the questions asked in the methods are answered. however, i did see that this format is the journal preference.
2. The reference 22 is incorrect- they did not use ICI with DT. Please consider replacing with Hamidi S, Iyer PC, Dadu R, Gule-Monroe MK, Maniakas A, Zafereo ME, Wang JR, Busaidy NL, Cabanillas ME. Checkpoint Inhibition in Addition to Dabrafenib/Trametinib for BRAFV600E-Mutated Anaplastic Thyroid Carcinoma. *Thyroid*. 2024 Mar;34(3):336-346. doi: 10.1089/thy.2023.0573. Epub 2024 Feb 13. PMID: 38226606.
3. They mention patients with the BRAF mutation were included in the study. It would be nice to know in the discussion- how long were these patients on BRAF directed therapy before they were started on Durvalumab and tremelimumab.
4. They focused on outcomes in patients with liver mets. Have they seen a difference in outcomes in patients with brain mets veruss those without?
5. They mentioned patients with DTC and MTC that progressed. what treatments were they transitioned to once they were off study?
6. Also they included PFS data for DTC and MTC- given the meaningful end point of OS with ATC, and the fact that their 18 months PFS was supposed to be studied per methods- they should report the ATC patients PFS esp if it was less than 18 months.
7. Line 455- states"baseline levels of key biomarkers in tumour 455 tissue and blood samples were correlated with efficacy outcomes" - what and how was this done- what biomarkers were used and these have not been described in results- please mention these.
8. Line 484 states: "Assuming a historical 6- month OS rate of 5% and an anticipated improvement to 35% (one-sided .=5%, 485 power=80%), 12 patients were needed." - what reference was used to show an improvement to 35%- please cite this

Reviewer #2

(Remarks to the Author)

This phase II, single-arm trial evaluated whether dual PD-L1/CTLA-4 blockade with durvalumab and tremelimumab could improve clinical outcomes in advanced thyroid cancers. Patients with DTC (n = 37) and MTC (n = 30) and progressive metastatic disease under prior standard systemic therapy were included, along with 12 ATC patients enrolled regardless of previous therapy.

Given the extremely poor prognosis of ATC and the limited efficacy of currently available systemic treatments, the outcomes observed in the ATC cohort are particularly encouraging, with a 6-month OS rate of 65.6% and an overall response rate of 33%. However, efficacy in the DTC and MTC cohorts was modest, with 6-month PFS rates of 38.8% and 35.1%, respectively.

As few studies have assessed the efficacy of immune checkpoint inhibitors (ICIs), such as anti-PD-1 antibodies, in thyroid cancers, this trial is of particular interest. An OS of 13.8 months and an 18-month survival rate of 46.9% highlight a clinically meaningful benefit of ICIs on overall survival in patients with ATC. The efficacy of the study treatment in the DTC and MTC cohorts remained modest, with 6-month PFS rates of 38.8% and 35.1%, respectively, as appropriately acknowledged in the discussion section.

Major Comments

- ATC patients were included regardless of BRAF status and prior therapy. As the ATA guidelines recommend: “In BRAFV600E-mutated stage IVC and unresectable stage IVB ATC patients who decline radiation therapy, initiation of BRAF/MEK inhibitors (dabrafenib plus trametinib) is recommended over other systemic therapies if available.” Please specify the BRAF status of the ATC patients included. For BRAFV600E-mutated ATC cases, justify management outside the scope of ATA recommendations.
- Specify the progression interval before inclusion for the DTC and MTC cohorts.
- The authors stated: “Durvalumab plus tremelimumab treatment in patients with ATC met the primary endpoint of this study, showing encouraging survival outcomes.” Please clarify in the Methods section what “meeting the primary endpoint” precisely means.

Minor Comments

- “Patients received durvalumab 1500 mg intravenously (IV) every four weeks plus tremelimumab 75 mg IV every four weeks for up to four cycles, followed by durvalumab monotherapy every four weeks.” Please justify the choice of these dosages and regimen.
- Are any pharmacokinetic or pharmacodynamic data available?
- Among patients who died during the study, the cause of death was not specified for 6 of 49 cases (12%). This proportion appears high for a prospective study involving close follow-up of patients with advanced cancer.
- Limited information is provided regarding tumor mutational burden, CPS scores, or TPS scores. Are these data available?
- Few data are presented on molecular alterations in the included cancers. Consider adding a table (possibly as supplementary material) listing key genetic drivers identified across cancer types (e.g., BRAF, RAS mutations, gene fusions, etc.). Please also specify the type of genetic testing performed (presumably NGS?).
- The authors mention RET fusions in MTC. As this event is exceptionally rare in MTC, please confirm whether any MTC cases harbored a RET fusion and specify the fusion type identified.
- Consider adding the following clarification in the Methods section: “Patients with locally advanced or metastatic ATC were included regardless of prior therapy.”
- Was there a centralized pathological review of tumor samples?
- From what percentage of a poorly differentiated component was a cancer classified as “poorly differentiated”?
- If correctly understood, two of the ten ATC patients underwent total thyroidectomy prior to inclusion. Histologically, were these tumors fully undifferentiated, or did they contain an undifferentiated component within a poorly or well-differentiated carcinoma?
- Can the authors provide MTC grading according to the publication J Clin Oncol. 2021 Nov 3;40(1):96–104. doi:10.1200/JCO.21.01329?
- The authors stated: “Previous studies on durvalumab (anti-PD-L1) and tremelimumab (anti-CTLA-4) for the treatment of neuroendocrine neoplasms yielded modest but clinically meaningful survival outcomes in a subset of patients with poorly differentiated carcinomas, along with a manageable safety profile.^{20,21}” Reference 21 does not appear to concern neuroendocrine tumors.
- The captions of Figure 3 regarding RET, PD-L1, BRAF, and MSI status are misaligned.
- The authors stated: “The combination of an anti-PD-1 agent with BRAF/MEK inhibitors or other MKIs yields objective responses in patients with ATC, often with a manageable safety profile.^{22,25}” Reference 22 does not seem to support this statement.
- The authors stated: “In this study, durvalumab plus tremelimumab showed modest efficacy in the MTC cohort, with survival outcomes inferior to those previously reported for MKIs.^{8,9,31}” However, in reference 31, all MTC patients were treated with selpercatinib as first-line therapy, and in references 8 and 9, 61% and 78% of patients, respectively, received vandetanib or cabozantinib as first-line therapy. Please provide bibliographic references of therapeutic trials conducted in pre-treated MTC patients, as in your study, for an appropriate comparison, as far as possible.
- In Table 1, check the percentages (e.g., for well-differentiated/poorly differentiated/unknown categories, and for ATC where totals exceed 100% [100 + 8.3%]; for MTC, 26.3 + 5.3 + 76.7 > 100%).
- Clarify in Table 1 whether “previous surgery” refers to total thyroidectomy or another type of surgical intervention.
- Consider including the median duration of response in Table 2 to provide a concise overview of key efficacy results.

- Did any ATC patients receive neck irradiation prior to inclusion?

Reviewer #3

(Remarks to the Author)

This is the statistical review for a single-arm trial with 3 cohorts (DTC/MTC/ATC) aiming to demonstrate efficacy based on 6-months PFS/OS rates. Only for the ATC cohort, the primary endpoint was successfully demonstrated. The MTC arm stopped recruitment prematurely.

There are several aspects in the description of methods and presentation of results that need to be addressed.

There is a discrepancy between how the trial was planned and how the primary endpoint is reported. The original sample size estimation was based on Simon's two stage design for binomial endpoints for DTC/MTC arms, and as single stage design for ATC with binomial endpoint, using 6-months PFS (DTC/MTC) and 6-months OS (ATC) rates. Primary endpoints are reported based on KM estimates, i.e. for time-to-event endpoints accounting for censoring. While I agree with using KM estimates, this discrepancy should be pointed out.

Please provide not only the percentage but also the absolute number of patients with PD/Death for the primary endpoints for all cohorts, including the number of censored patients.

Please provide a reference for the assumption of a 6-month OS rate of 5% for the ATC arm. I could not find those in the manuscript or study protocol.

Please include the number of pts needed to be alive after 6 months out of 12 patients for the single stage design in cohort 3 according to the sample size planning.

The primary endpoint is planned for a one-sided alpha of 5%, so the statement 'All statistical tests used for efficacy and safety analyses were two-tailed and results with p-values <0.05 were considered significant' is not correct.

RECIST/ Suppl table 1: According to the study protocol, RECIST 1.1 will be used for response assessment. I think, this should be the reason why the primary endpoints are reported based on RECIST, not the strong agreement with irRECIST 1.1.

In the discussion, the authors state 'exploratory biomarker analyses were performed to identify potential predictors of the response. These results suggest that emerging biomarkers, including PD-L1 expression and MSI status, may help refine patient selection.' Apart from the annotation shown in figure 3 I could not find any analysis on these data, or any analysis supporting this statement. In fact, biomarker is rather scarce and not done for the majority of patients.

Agreement analysis. A significant kappa > 0 is not the same as 'No significant differences were observed between the two evaluation methods'. Please rephrase this.

The Clopper-Pearson method for CIs is not adequate for censored data.

Please specify the date of data cut-off for the analysis.

Results on time-dependent analysis of imAEs (line 182/183): Please correct, it is the occurrence of imAEs and not the endpoint PFS/OS which is analysed as time-dependent factor.

Figure 3a/b waterfall plots: numbers do not add up, it should be 32 DTC patients etc but it is less.

Version 1:

Reviewer comments:

Reviewer #1

(Remarks to the Author)

This is a phase II, single-arm, multi-cohort trial was conducted determine whether dual anti-PD-L1/CTLA-4 inhibition using durvalumab and tremelimumab can improve the clinical outcomes in advanced thyroid cancer.

This is a much needed, invaluable study for advanced thyroid cancer and makes a tremendous contribution to what we have as treatment options for some of the aggressive thyroid cancers.

Thank you for clarifying the comments and suggestions. Given that a third of patients had received radiation before the drug combination, did you see any difference in survival of those who received radiation to neck before durval and treme combo versus who did not? Did any of these undergo surgery due to tumor shrinkage?

Also one of the comments by author is that they didnt see any benefit of adding a second ICI over single agent. I dont think this is necessarily true as better results were seen from an OS standpoint with ipilimumab and nivolumab.

Overall, the paper looks ready for publication if they could add a comment of if there was any difference in survival in

between those who received radiation to neck versus not.

Reviewer #3

(Remarks to the Author)

All my points have been addressed by the authors.

One minor issue: In the new suppl. tables 1 and 2, the table head should probably read 'estimated cumulative survival rate' instead of 'estimated cumulative survival ratio'

AUTHORS' RESPONSE TO REVIEWERS

Reviewer 1: This is a very interesting phase II, single-arm, multi-cohort trial was conducted to 70 determine whether dual anti-PD-L1/CTLA-4 inhibition using durvalumab and tremelimumab can improve the clinical outcomes in advanced thyroid cancer.

The results of this trial are specifically meaningful to the ATC cohort of patients particularly the BRAF wild type ones for who no meaningful drugs are available.

Authors' response: The authors want to sincerely thank reviewer 1 for the time and dedication invested in reviewing our manuscript. We deeply value your comments and especially appreciate your recognition of the study's interest and the clinical relevance of the results obtained, particularly for the cohort of patients with BRAF wild-type anaplastic thyroid carcinoma (ATC), who currently lack effective therapeutic options and performed quite well in this study. Your comments motivate us to continue research in this area with the goal of offering meaningful therapeutic alternatives for this patient population with unmet medical needs.

Please accept our warmest regards and, once again, our thanks for your valuable contribution.

Reviewer 1: I have provided specific questions in the attached edited version of manuscript but the main points to focus on are:

1. The methods state that 18 month PFS would be reported for ATC but the results is missing that. Honestly, having methods above results is preferred mainly because one can refer results to methods and make sure the questions asked in the methods are answered. However, I did see that this format is the journal preference.

Authors' response: The 18-month rate for PFS and OS are placed in Table 2. This table summarizes the main efficacy endpoints of our study. Following your request further information is given in the main text. Thank you so much.

Reviewer 1:

2. The reference 22 is incorrect- they did not use ICI with DT. Please consider replacing with Hamidi S, Iyer PC, Dadu R, Gule-Monroe MK, Maniakas A, Zafereo ME, Wang JR, Busaidy NL, Cabanillas ME. Checkpoint Inhibition in Addition to Dabrafenib/Trametinib for BRAFV600E-Mutated Anaplastic Thyroid Carcinoma. *Thyroid*. 2024 Mar;34(3):336-346. doi: 10.1089/thy.2023.0573. Epub 2024 Feb 13. PMID: 38226606.

Authors' response: Thank you for your comment and for providing this valuable citation. The reference has been replaced.

Reviewer 1:

3. They mention patients with the BRAF mutation were included in the study. It would be nice to know in the discussion- how long were these patients on BRAF directed therapy before they were started on Durvalumab and tremelimumab.

Authors' response: We agree that this information would be interesting. Among the 6 patients with known BRAF mutated status, two received dabrafenib plus trametinib for a week and 11 months. Some information has been added following your request to results and discussion sections:

BRAF/MEK inhibitors were administered prior to two patients with ATC, one for a duration of one week and another for 11 months. Durvalumab and tremelimumab provided in these two patients a PFS interval of 3 and 10 months after BRAF/MEK inhibitors, respectively.

The two patients with ATC who received prior BRAF/MEK inhibitors showed promising PFS with durvalumab and tremelimumab, suggesting ICI is yet a valuable option after failure of TKIs in these patients.

Reviewer 1:

4. They focused on outcomes in patients with liver mets. Have they seen a difference in outcomes in patients with brain mets versus those without?

Authors' response: Brain metastases are always of interest and are related with worse prognosis. However, taking into account the bad prognosis of the patients with ATC and the patients with DTC and MTC who progressed to 2 prior systemic treatments, and the inclusion criteria of a life expectancy >3 months, patients with brain metastasis were probably excluded. Additionally, brain metastases are not so common among patients with thyroid cancers. Only one patient, with follicular subtype, had a target or non target lesion located in the brain.

For the study of immune checkpoint inhibitors, liver metastases are of interest because the liver mostly has a "cold" immune microenvironment that limits the expected effect of ICI through the increased immune surveillance. The higher rate of CD8+ regulatory lymphocytes in the liver help to reduce the potential immune-mediated antitumoral response evoked by ICIs .

Reviewer 1:

5. They mentioned patients with DTC and MTC that progressed. what treatments were they transitioned to once they were off study?

Authors' response: These patients were followed until study closure and all the subsequent treatments are listed on Supplementary Table 2:

	DTC n=3	MTC n=4	ATC n=2
Blinded clinical trial	1 (33%)	1 (25%)	-
Cabozantinib	1 (33%)	1 (25%)	-
Lenvatinib	2 (67%)	-	-
Selpercatinib	-	2 (50%)	-
Carboplatin	-	-	1 (50%)
Taxol	1 (33%)	-	1 (50%)
Gemcitabine	-	-	1 (50%)
Pegylated liposomal doxorubicin	-	-	1 (50%)
Radiotherapy	1 (33%)	-	-
No subsequent treatment	-	-	1 (50%)

To make more clear the availability of this information, we modified the following sentence:

The details of all treatment regimens administered to the patients after progression to the study treatment are shown in Supplementary Table 4.

Reviewer 1:

6. Also they included PFS data for DTC and MTC- given the meaningful end point of OS with ATC, and the fact that their 18 months PFS was supposed to be studied per methods- they should report the ATC patients PFS esp if it was less than 18 months.

Authors' response: We agree that the PFS for patients with ATC should be reported. The median PFS for patients with ATC is placed in Table 2 and, following your suggestion, is given in the main text.

Reviewer 1:

7. Line 455- states"baseline levels of key biomarkers in tumour 455 tissue and blood samples were correlated with efficacy outcomes" - what and how was this done- what biomarkers were used and these have not been described in results- please mention these.

Authors' response: Thank you for your comment. We acknowledge that this sentence is too vague. We deleted the phrase because it is unnecessary, we explained the exploratory analyses in the section "Molecular Assessments". We included correlative analysis of PD-L1 expression, presence of MSI and RET fusions with the efficacy outcomes. Results are reported within the manuscript.

Reviewer 1:

8. Line 484 states: "Assuming a historical 6- month OS rate of 5% and an anticipated improvement to 35% (one-sided $\alpha=5\%$, 485 power=80%), 12 patients were needed." - what reference was used to show an improvement to 35%- please cite this.

Authors' response: This is extrapolated from the standard treatment practice. Classical studies with chemotherapy reported dismal OS, with 6-months OS rates of approximately 5% ¹⁻². We may agree that an improvement to 35% would be considered clinically meaningful as $\frac{1}{3}$ of patients will increase their life expectancy regarding other standard options and more in such a

pretreated population. Spartalizumab showed an OS rate of 40% at one year in ATCs ³.

Refs:

1. Shimaoka K, Schoenfeld DA, DeWys WD, Creech RH, DeConti R. A randomized trial of doxorubicin versus doxorubicin plus cisplatin in patients with advanced thyroid carcinoma. *Cancer*. November 1, 1985;56(9):2155–2160.
2. Sosa JA, Elisei R, Jarzab B, Balkissoon J, Lu S-P, Bal C, et al. Randomized safety and efficacy study of fosbretabulin with paclitaxel/carboplatin against anaplastic thyroid carcinoma. *Thyroid Off J Am Thyroid Assoc*. February 2014;24(2):232–240.
3. Capdevila J, Wirth LJ, Ernst T, Ponce Aix S, Lin C-C, Ramlau R, et al. PD-1 Blockade in Anaplastic Thyroid Carcinoma. *J Clin Oncol Off J Am Soc Clin Oncol*. August 10, 2020;38(23):2620–2627.

Reviewer 1. Questions from the attached document:

The following questions are comments placed on the document attached *reviewer_1_attachment_1760599799_41*.

Reviewer 1 (lines 105-107): please cite references- spartalizumab study, keynote for single agent pembrolizumab.

Authors' response: Thank you for your suggestion. These two references have been included:

1. Oh, D.-Y. et al. Efficacy and safety of pembrolizumab monotherapy in patients with advanced thyroid cancer in the phase 2 KEYNOTE-158 study. *Cancer* 129, 1195–1204 (2023).
2. Capdevila, J. et al. PD-1 Blockade in Anaplastic Thyroid Carcinoma. *J Clin Oncol* 38, 2620–2627 (2020).

Reviewer 1 (line 121): the methods section is missing

Authors' response: as per Nature journal guidelines, the methods are placed after the discussion.

Reviewer 1 (line 132): what were the eligibility criteria? Why was the PFS for ATC not studied or mentioned in methods or results?

Authors' response: Eligibility criteria are mentioned in the Methods section. We focused on overall survival for the ATC cohort. However, the PFS for ATC was also reported in Table 2:

Outcomes	DTC n= 37	MTC n = 30	ATC n = 12
Median PFS; months (95% CI)	3.7 (2.7-6.5)	5.3 (2.8-23.2)	3.6 (2.2-NR)
6-month PFS rate; % (95% CI)	32.4 (20.4-51.6)	40.9 (26.3-63.6)	33.3 (15.0-74.2)
18-month PFS rate; % (95% CI)	6.0 (1.6-22.6)	26.5 (13.6-51.6)	8.3 (1.3-54.4)
Median OS; months (95% CI)	13.8 (9.4-32.8)	35.5 (30.1-NR)	13.8 (5.7-NR)
6-month OS rate; % (95% CI)	70.3 (56.7-86.7)	96.6 (90.1-100)	65.6 (43.2-99.8)
18-month OS rate; % (95% CI)	41.5 (28.0-61.4)	78.8 (65.3-97.5)	46.9 (25.0-87.9)

Reviewer 1 (line 139): the methods states that 18 month PFS were to be reported but its not mentioned here

Authors' response: We reported in the text the main endpoints. The 18-month rate for PFS and OS are placed in Table 2 because we believe that is more helpful for the readers to find all endpoints summarized in a single place.

Reviewer 1 (line 163): what treatment was offered when they progressed?

Authors' response: The subsequent treatments are listed in Supplementary Table 2. To make more clear the availability of this information, we modified the following sentence:

The details of all treatment regimens administered to the patients after progression to the study treatment are shown in Supplementary Table 2.

Reviewer 1 (line 176): why liver metastases were the focus here? Why not brain mets?

Authors' response: Despite brain metastases being always of interest, they were present only in one patient. Liver metastases are a known factor that can reduce the efficacy of ICIs, making it of special interest to study.

Reviewer 1 (line 195): Did they receive BRAF directed therapy before getting on this? how long were they on BRAF therapy

Authors' response: We agree that this information would be interesting. Further information has been added. No responses were reported among patients receiving previous BRAF/MEK inhibitors

Reviewer 1 (line 200): at what point in their trial did they withdraw consent. Was data censored to the date of withdrawal?

Authors' response: Reviewing the data, we noticed that the reason for the end of treatment was the patient's decision. One patient was followed until study closure and the other two withdrew their consent. This data has been clarified in the manuscript:

The reasons for treatment discontinuation included disease progression (54 patients [68.4%]), unacceptable toxicity (seven patients [8.9%]), death (six patients [7.6%]), patient's decision withdrawal of consent (three patients [3.8%], two of them withdraw the consent), and loss to follow-up (one patient [1.3%]).

For the patients who withdrew their consent, the data was censored to the date of withdrawal. One patient decided to finish the treatment before the second cycle and the two others at cycle 10 and 20 with durvalumab.

Reviewer 1: please cite hamidit et al. checkpoint inhibitor combined with anti-PD1. ref 22 is incorrect- they didn't use ICI

Authors' response: Thank you for your comment and for providing this valuable citation. The reference has been replaced.

Reviewer 1 (line 268): why?

Authors' response: *Due to the better efficacy shown with dual ICI over single ICI in several solid tumours, dual ICI were substituting single ICI treatments. However, our results do not support the addition of the second ICI suggesting that single therapeutic strategies may provide similar efficacy with lower toxicity and economic cost.*

Reviewer 1 (line 370): this needs to be moved above the results.

Authors' response: *Per guidelines, the journals belonging to the Nature family put the Methods section after Discussion.*

Reviewer 1 (line 428): why not ATC?

Authors' response: *For DTC and MTC there were previous investigations providing rationale for the use of PFS. However, in patients with ATC, due to the aggressive nature of the disease, with no standard therapy showing benefit, and an expected OS of 3-4 months, 6-month OS rate could provide better evidence of a potential benefit.*

Reviewer 1 (line 442): this is not reported in results

Authors' response: *The 18-month rate for PFS and OS are placed in Table 2 and now added within the main text in the results section of the manuscript. Table 2 summarizes the main efficacy endpoints of our study.*

Reviewer 1 (line 455): what were the biomarkers?

Authors' response: *Thank you for your comment. We acknowledge that this sentence is too vague. We deleted the sentence because we explained the exploratory analyses in the section "Molecular Assessments".*

Reviewer 1 (line 485): please cite reference

Authors' response: As explained, based on previous experience with anti PD-1 and the benchmark studies with chemotherapy reporting dismal OS rates, we consider that an increase in 30% would be of clinical relevance.

Reviewer 2: This phase II, single-arm trial evaluated whether dual PD-L1/CTLA-4 blockade with durvalumab and tremelimumab could improve clinical outcomes in advanced thyroid cancers. Patients with DTC (n = 37) and MTC (n = 30) and progressive metastatic disease under prior standard systemic therapy were included, along with 12 ATC patients enrolled regardless of previous therapy.

Given the extremely poor prognosis of ATC and the limited efficacy of currently available systemic treatments, the outcomes observed in the ATC cohort are particularly encouraging, with a 6-month OS rate of 65.6% and an overall response rate of 33%. However, efficacy in the DTC and MTC cohorts was modest, with 6-month PFS rates of 38.8% and 35.1%, respectively.

As few studies have assessed the efficacy of immune checkpoint inhibitors (ICIs), such as anti-PD-1 antibodies, in thyroid cancers, this trial is of particular interest. An OS of 13.8 months and an 18-month survival rate of 46.9% highlight a clinically meaningful benefit of ICIs on overall survival in patients with ATC. The efficacy of the study treatment in the DTC and MTC cohorts remained modest, with 6-month PFS rates of 38.8% and 35.1%, respectively, as appropriately acknowledged in the discussion section.

Authors' response: The authors sincerely appreciate your valuable comments and feedback on our manuscript. We are pleased to learn that you find this phase II study of the combination of dual PD-L1/CTLA-4 blockade with durvalumab and tremelimumab in advanced thyroid cancers of interest.

We fully agree that the results observed in the anaplastic thyroid carcinoma (ATC) cohort are encouraging, given the extremely poor prognosis of this entity and the limited therapeutic options available. The 6-month overall

survival (OS) rate of 65.6% and the objective response rate of 33% suggest a potentially significant clinical benefit in this subgroup of patients. We appreciate your also highlighting the 18-month survival rate of 46.9%, which we believe reinforces the clinical relevance of this treatment in ATC.

Regarding the differentiated (DTC) and medullary carcinoma (MTC) cohorts, we share your assessment that efficacy was more modest, as clearly noted in our discussion. Nevertheless, we believe these findings are still informative, considering that the included patients had progressive metastatic disease under prior standard systemic therapy, representing a complex clinical scenario with unmet medical need. Furthermore, the fact that this study includes a relatively representative number of patients with DTC and MTC treated with ICIs in a prospective setting contributes to the currently limited body of evidence in this field.

One of the central motivations of this study was precisely to explore the potential benefit of immunotherapy in thyroid cancer subtypes that have been little investigated so far. Although the results in DTC and MTC do not reach the magnitude observed in ATC, we believe they provide relevant information on the clinical efficacy of this therapeutic strategy, which may guide future research, including the development of combined strategies or more personalized approaches.

We once again appreciate your constructive comments, which have allowed us to reach more deeply on the interpretation of our results and their relevance in the current clinical context.

Reviewer 2: Major Comments

- ATC patients were included regardless of BRAF status and prior therapy. As the ATA guidelines recommend: “In BRAFV600E-mutated stage IVC and unresectable stage IVB ATC patients who decline radiation therapy, initiation of BRAF/MEK inhibitors (dabrafenib plus trametinib) is recommended over other systemic therapies if available.”

Please specify the BRAF status of the ATC patients included. For BRAFV600E-mutated ATC cases, justify management outside the scope of ATA recommendations.

Authors' response: Our study was designed following the guidelines and regulations applicable in Spain. The use of BRAF/MEK inhibitors in ATCs is currently not approved either by the European Medicines Agency or the Spanish Agency of Medicines and Medical Devices. This situation highlights the importance of our study in this orphan population. Additionally, BRAF mutation status was not assessed routinely in patients with advanced thyroid carcinomas. This information is only available in a few cases, as shown in Figure 3.

Reviewer 2:

- Specify the progression interval before inclusion for the DTC and MTC cohorts.

Authors' response: The median progression interval before inclusion has been added to the Table 1:

Characteristics	DTC n= 37	MTC n = 30	ATC n = 12	GETNE-DUTHY n = 79
Time from last progression prior study inclusion				
Median; months (95% CI)	0.8 (0.7-1.1)	0.9 (0.5-1.3)	0.8 (NA)	0.8 (0.7-1.1)

Reviewer 2:

- The authors stated: “Durvalumab plus tremelimumab treatment in patients with ATC met the primary endpoint of this study, showing encouraging survival outcomes.”

Please clarify in the Methods section what “meeting the primary endpoint” precisely means.

Authors' response: Thank you for your suggestion, we agree that this clarification would improve the understanding of our results. We added the following fragment to our text:

For the DTC and MTC cohorts, the primary endpoint was considered met if a 6-months PFS rate of 45% was achieved. For the ATC cohort, the primary endpoint was considered met if a 6-months OS of 35% was achieved.

Reviewer 2: Minor Comments

- “Patients received durvalumab 1500 mg intravenously (IV) every four weeks plus tremelimumab 75 mg IV every four weeks for up to four cycles, followed by durvalumab monotherapy every four weeks.”

Please justify the choice of these dosages and regimen.

Authors’ response: Thank you for your suggestion. We added this information in Methods:

The dosages were selected using pharmacokinetic models developed with data from a phase 1 study for durvalumab ¹, and phase 1 to 3 for tremelimumab ². The rationale for the selected scheme was provided by previous clinical trials with dual ICI in patients with advanced melanoma and phase 1 trials with durvalumab plus tremelimumab ³.

1. Baverel PG, Dubois VFS, Jin CY, et al: Population Pharmacokinetics of Durvalumab in Cancer Patients and Association With Longitudinal Biomarkers of Disease Status. Clin Pharmacol Ther 103:631–642, 2018
2. Wang E, Kang D, Bae K-S, et al: Population pharmacokinetic and pharmacodynamic analysis of tremelimumab in patients with metastatic melanoma. J Clin Pharmacol 54:1108–1116, 2014
3. Antonia S, Goldberg SB, Balmanoukian A, et al: Safety and antitumour activity of durvalumab plus tremelimumab in non-small cell lung cancer: a multicentre, phase 1b study. Lancet Oncol 17:299–308, 2016

Reviewer 2:

- Are any pharmacokinetic or pharmacodynamic data available?

Authors’ response: No pharmacokinetic analysis was conducted in our study. The Pharmacokinetics of durvalumab and tremelimumab was already characterized in previous studies and is available in the Investigator Brochure. PKs is a classical endpoint for phase I trials and this was not in the scope of

this phase II study. You may also count with further information in the following references:

- Baverel PG, Dubois VFS, Jin CY, et al: Population Pharmacokinetics of Durvalumab in Cancer Patients and Association With Longitudinal Biomarkers of Disease Status. Clin Pharmacol Ther 103:631–642, 2018
- Wang E, Kang D, Bae K-S, et al: Population pharmacokinetic and pharmacodynamic analysis of tremelimumab in patients with metastatic melanoma. J Clin Pharmacol 54:1108–1116, 2014

The study analysed the presence of several biomarkers (PD-L1 Positivity, MSI and presence of RET genetic alterations) and correlated them with efficacy endpoints. This was an exploratory objective. The genetic alterations are shown in Figure 3.

We showed some results but due to the small number and the limited number of samples analyzed the interpretability of these exploratory endpoints had low impact. This has been shown as a constraint of the study in the discussion:

Fourth, patients who showed treatment response were included in the biomarker analyses; therefore, the possibility of post-hoc selection bias cannot be completely excluded. However, these analyses were exploratory, and the results were interpreted cautiously.

Reviewer 2:

- Among patients who died during the study, the cause of death was not specified for 6 of 49 cases (12%). This proportion appears high for a prospective study involving close follow-up of patients with advanced cancer.

Authors' response: Thank you for your comment. We revised the causes of exitus with the participant centers. Three of them died due to disease progression and other due to an adverse event not related to treatment. The other two patients died at home and unfortunately were lost to follow-up. The COVID-19 pandemic possibly diffculted a complete follow-up of these patients. This information has been updated in the manuscript.

Reviewer 2:

- Limited information is provided regarding tumor mutational burden, CPS scores, or TPS scores. Are these data available?

Authors' response: The data available is limited. We introduced the information as Supplementary table 3 and already discussed this as a limiting factor in the discussion. This was an investigator-initiated study and the budget for translational research was very limited, precluding the extensive molecular profiling of all patients.

Reviewer 2:

- Few data are presented on molecular alterations in the included cancers. Consider adding a table (possibly as supplementary material) listing key genetic drivers identified across cancer types (e.g., BRAF, RAS mutations, gene fusions, etc.). Please also specify the type of genetic testing performed (presumably NGS?).

Authors' response: Please, consult the new table added as a Supplementary Table 3. The techniques used are also disclosed in the methods section, molecular analysis subsection.

Reviewer 2:

- The authors mention RET fusions in MTC. As this event is exceptionally rare in MTC, please confirm whether any MTC cases harbored a RET fusion and specify the fusion type identified.

Authors' response: Thank you for your comment. In patients with DTC only RET fusions were detected, while in patients with MTC there were detected only RET mutations. No RET fusions are detected in MTCs, only in DTCs. RET mutations occur in an elevated percentage of patients with MTC (above 50%), specially in advanced disease in which some studies reported a prevalence of the mutation of up to 80%. These RET mutations were detected in eleven (73.3%) patients with MTCs. RET inhibitors such as selipratinib and pralsetinib are viable treatment options for these patients. The available

information has been added to Supplementary Table 5. The manuscript has been revised and the following texts have been corrected for more clarity:

In Efficacy Results:

RET status was locally assessed in 15 patients with MTC (50.0%), of whom 11 (73.3%) had mutations ~~or fusions~~. Data on RET status were available for one patient with MTC who achieved an objective response, and the data showed no RET mutations ~~or fusions~~.

In Material and Methods:

The RET M918T mutation in patients with MTC and RET fusions in patients with DTC were assessed locally using NGS.

Reviewer 2:

- Consider adding the following clarification in the Methods section: “Patients with locally advanced or metastatic ATC were included regardless of prior therapy.”

Authors’ response: Following your suggestion, the cohort definitions were modified for more clarity:

Patients with confirmed locally advanced or metastatic thyroid carcinoma were recruited into the following three cohorts: [...], and patients with ATC regardless of prior therapy ~~whether they previously underwent therapy or not~~ (Cohort 3).

Reviewer 2:

- Was there a centralized pathological review of tumor samples?

Authors’ response: The diagnostic and pathological review was assessed locally by the investigators. The reason to not perform the central review was budget constraints. Only a few exploratory analyses were planned. We acknowledge that a central review would have increased the validity of our study but it should be remarked that all study centers were reference hospitals

in Spain with highly qualified pathologist and multidisciplinary collaborative teams for the diagnosis and management of the patients.

Reviewer 2:

- From what percentage of a poorly differentiated component was a cancer classified as “poorly differentiated”?

Authors’ response: The pathological review was locally performed by experienced pathologists from reference centers following the current clinical guidelines for the differential diagnosis of thyroid cancer. It was confirmed with the centers that poorly differentiated cancer was a variant of DTC.

Reviewer 2:

- If correctly understood, two of the ten ATC patients underwent total thyroidectomy prior to inclusion. Histologically, were these tumors fully undifferentiated, or did they contain an undifferentiated component within a poorly or well-differentiated carcinoma?

Authors’ response: Two of the ten patients with ATC underwent thyroidectomy before study inclusion. The surgery outcome was R0 for one and R2 for the other. All patients with ATC were undifferentiated according to the pathological diagnosis from the reference center.

Reviewer 2:

- Can the authors provide MTC grading according to the publication J Clin Oncol. 2021 Nov 3;40(1):96–104. doi:10.1200/JCO.21.01329?

Authors’ response: Thank you for providing us with this interesting publication. The sponsor agrees that reporting mitotic rate of Ki-67 might have been of interest. Nevertheless, at the time of study conception and conduction, the Ki-67 implementation was not established in MTC. There is no centralized sample revision and, therefore, this data cannot be obtained.

Reviewer 2:

- The authors stated: “Previous studies on durvalumab (anti-PD-L1) and tremelimumab (anti-CTLA-4) for the treatment of neuroendocrine neoplasms yielded modest but clinically meaningful survival outcomes in a subset of patients with poorly differentiated carcinomas, along with a manageable safety profile.^{20,21}”

Reference 21 does not appear to concern neuroendocrine tumors.

Authors’ response: Thank you for the comment. The reference 21 was incorrectly added and has been deleted.

Reviewer 2:

- The captions of Figure 3 regarding RET, PD-L1, BRAF, and MSI status are misaligned.

Authors’ response: Thank you for your observation. The figure has been replaced with a new one with the captions correctly aligned.

Reviewer 2:

- The authors stated: “The combination of an anti-PD-1 agent with BRAF/MEK inhibitors or other MKIs yields objective responses in patients with ATC, often with a manageable safety profile.^{22,25}”

Reference 22 does not seem to support this statement.

Authors’ response: The reference 22 has been replaced by the following one:

Hamidi, S. et al. Checkpoint Inhibition in Addition to Dabrafenib/Trametinib for BRAFV600E-Mutated Anaplastic Thyroid Carcinoma. Thyroid 34, 336–346 (2024).

Reviewer 2:

- The authors stated: “In this study, durvalumab plus tremelimumab showed modest efficacy in the MTC cohort, with survival outcomes inferior to those previously reported for MKIs.^{8,9,31}” However, in reference 31, all MTC patients were treated with seliperatinib as first-line therapy, and in references 8 and 9, 61% and 78% of patients, respectively, received vandetanib or cabozantinib as first-line therapy. Please provide bibliographic references of therapeutic

trials conducted in pre-treated MTC patients, as in your study, for an appropriate comparison, as far as possible.

Authors' response: Thank you for your comment. The references have been revised and replaced by the appropriate ones. In the new references, efficacy outcomes for pre-treated MTC patients who received selpercatinib, cabozantinib, vandetanib and pralsetinib are provided:

1. Kreissl, M. C. et al. Efficacy and Safety of Vandetanib in Progressive and Symptomatic Medullary Thyroid Cancer: Post Hoc Analysis From the ZETA Trial. *J Clin Oncol* 38, 2773–2781 (2020).
2. Cabanillas ME, de Souza JA, Geyer S, Wirth LJ, Menefee ME, Liu SV, et al. Cabozantinib As Salvage Therapy for Patients With Tyrosine Kinase Inhibitor-Refractory Differentiated Thyroid Cancer: Results of a Multicenter Phase II International Thyroid Oncology Group Trial. *J Clin Oncol Off J Am Soc Clin Oncol*. October 10, 2017;35(29):3315–3321.
3. Subbiah V, Hu MI, Mansfield AS, Taylor MH, Schuler M, Zhu VW, et al. Pralsetinib in Patients with Advanced/Metastatic Rearranged During Transfection (RET)-Altered Thyroid Cancer: Updated Efficacy and Safety Data from the ARROW Study. *Thyroid Off J Am Thyroid Assoc*. January 2024;34(1):26–40.
4. Wirth LJ, Sherman E, Robinson B, Solomon B, Kang H, Lorch J, et al. Efficacy of Selpercatinib in RET-Altered Thyroid Cancers. *N Engl J Med*. August 26, 2020;383(9):825–835.

Reviewer 2:

- In Table 1, check the percentages (e.g., for well-differentiated/poorly differentiated/unknown categories, and for ATC where totals exceed 100% [100 + 8.3%]; for MTC, 26.3 + 5.3 + 76.7 > 100%).

Authors' response: Apologies for this error. Table 1 has been revised.

Reviewer 2:

- Clarify in Table 1 whether “previous surgery” refers to total thyroidectomy or another type of surgical intervention.

Authors' response: Previous surgery includes partial or total thyroidectomy, but also a few cases of laryngectomy and chordectomy. This information has been included in Table 1.

Reviewer 2:

- Consider including the median duration of response in Table 2 to provide a concise overview of key efficacy results.

Authors' response: Following your suggestion, the median duration of response has been added to Table 2.

Reviewer 2:

- Did any ATC patients receive neck irradiation prior to inclusion?

Authors' response: Four patients with ATC received neck irradiation prior to inclusion. This information has been included in the text as follows:

In patients with ATC, 33.3% of them were treated with local radiotherapy prior to inclusion.

Reviewer 3: This is the statistical review for a single-arm trial with 3 cohorts (DTC/MTC/ATC) aiming to demonstrate efficacy based on 6-months PFS/OS rates. Only for the ATC cohort, the primary endpoint was successfully demonstrated.

The MTC arm stopped recruitment prematurely.

There are several aspects in the description of methods and presentation of results that need to be addressed.

There is a discrepancy between how the trial was planned and how the primary endpoint is reported. The original sample size estimation was based on Simon's two stage design for binomial endpoints for DTC/MTC arms, and as single stage design for ATC with binomial endpoint, using 6-months PFS (DTC/MTC) and 6-months OS (ATC) rates.

Primary endpoints are reported based on KM estimates, i.e. for time-to-event endpoints accounting for censoring. While I agree with using KM estimates, this discrepancy should be pointed out.

Authors' response: Thank you very much for your thoughtful and detailed review of the statistical aspects of the trial. The sponsor really appreciates your valuable feedback, that has contributed to improve the quality and accuracy of this study.

We appreciate your observation regarding the discrepancy between the originally planned analysis and the way the primary endpoints are reported. While the analysis was indeed performed using Kaplan-Meier (KM) estimates to appropriately account for censoring in the time-to-event data, we acknowledge that this represents a deviation from the initial binomial framework described in the design. The use of Kaplan-Meier method is widely accepted to address survival rates at several timepoints and it is more consistent when tumor assessments for all patients are not exactly performed within the estimated endpoints, which was the case. Some CT-scans were performed within a time window from the 6 months, so the sponsor decided that Kaplan-Meier estimates would give a more accurate estimation. All patients were assessed within +/-1 month after 6 months of treatment initiation, so this limitation has a really low impact when considering Kaplan-Meier estimates. The sponsor agrees that this methodological change should be clearly explained and justified in the manuscript to ensure transparency and consistency. The methods section has been reviewed to explicitly describe this point, clarify the rationale for reporting KM estimates, and discuss any implications for the interpretation of the findings:

Although the protocol planned the estimation of primary endpoints using binomial calculations, some tumor assessments and survival follow ups were performed within +/- 1 month from the 6 month expected time point due to changes in scheduled visits per patient or site availability, or different clinical courses of the disease, so the sponsor decided that Kaplan Meier estimated would give a more accurate estimation at 6 months. Binomial calculations are listed in Supplementary table 8.

Reviewer 3: Please provide not only the percentage but also the absolute number of patients with PD/Death for the primary endpoints for all cohorts, including the number of censored patients.

Authors' response: *Thank you for your suggestion. The information has been added as Supplementary Tables 1 and 2.*

Reviewer 3: Please provide a reference for the assumption of a 6-month OS rate of 5% for the ATC arm. I could not find those in the manuscript or study protocol.

Authors' response: *This is extrapolated from the standard treatment practice. Following your request, the Sponsor has added a reference to benchmark studies for the use of chemotherapy that classically is used for ATCs, despite it should be considered that these studies were performed exclusively in the first treatment line setting whereas our study allowed the inclusion of patients after progression to standard treatment options. Pretreated patients tend to have worse prognosis than treatment naive. This was taken into account for study design and OS rate assumptions. In our cohort, 42% of patients were pretreated with two prior treatment lines, which is correlated with dismal outcomes.*

Reference:

Shimaoka K, Schoenfeld DA, DeWys WD, Creech RH, DeConti R. A randomized trial of doxorubicin versus doxorubicin plus cisplatin in patients with advanced thyroid carcinoma. Cancer. November 1, 1985;56(9):2155–2160.

Reviewer 3: Please include the number of pts needed to be alive after 6 months out of 12 patients for the single stage design in cohort 3 according to the sample size planning.

Authors' response: *Following your request the number of patients required to reject null hypothesis is added in the statistical methods section:*

Assuming a historical 6-month OS rate of 5% and an anticipated improvement to 35% (one-sided $\alpha=5\%$, power=80%), 12 patients were needed and ≥ 3 patients alive at 6 months to declare the study positive.

Reviewer 3: The primary endpoint is planned for a one-sided alpha of 5%, so the statement 'All statistical tests used for efficacy and safety analyses were two-tailed and results with p-values <0.05 were considered significant' is not correct.

Authors' response: *Thank you so much for noticing. It is true that the primary endpoint tests were one-sided. The statistical methods section has been amended:*

Primary statistical tests were one-tailed and other statistical tests used for efficacy and safety analyses were two-tailed, and results with p-values <0.05 were considered significant.

Reviewer 3: RECIST/ Suppl table 1: According to the study protocol, RECIST 1.1 will be used for response assessment. I think this should be the reason why the primary endpoints are reported based on RECIST, not the strong agreement with irRECIST 1.1.

Authors' response: *The study was designed by 2019/2020 when immunotherapy use in thyroid cancers was scarce.*

RECIST 1.1, which was at that time more commonly used for the oncologists, was taken as the primary endpoint. That was the main reason to choose RECIST as primary criteria over irRECIST. However, the investigators assessed CT scans by both criteria to check consistency of the results. Our analysis indicates that RECIST and irRECIST were very concordant. We believe the use of irRECIST and the check of consistency between both criteria strengthened our results and increased validity.

Reviewer 3: In the discussion, the authors state 'exploratory biomarker analyses were performed to identify potential predictors of the response. These results suggest that emerging biomarkers, including PD-L1 expression and MSI status, may help refine patient selection.' Apart from the annotation shown in figure 3 I could not find any analysis on these data, or any analysis supporting this statement. In fact, biomarker is rather scarce and not done for the majority of patients.

Authors' response: The discussion also states that these exploratory analyses faced several constraints and limitations, mainly due to budget limitations which precluded the analysis of all samples. Additionally some patient samples could not be collected or the material was insufficient. The discussion states:

This study had several limitations. First, PD-L1 status was not systematically assessed in all patients; thus, we could not reach definitive conclusions regarding its predictive value. Future studies are needed to comprehensively evaluate the molecular biomarkers of response to ICIs in patients with thyroid cancer, particularly those with ATC.

The molecular results are reported in Figure 3 and were very exploratory due to the limitations. According to your observation, we also tempered the sentence commented here so it is very clear that our results could not reach to solid conclusions but rather point towards the need of additional translational research in this context:

Notably, exploratory biomarker analyses were performed with a subset of patients to identify potential predictors of the response. These exploratory preliminary results suggest that emerging biomarkers, including PD-L1 expression and MSI status, may help refine patient selection, despite further research will be needed.

Reviewer 3: Agreement analysis. A significant kappa > 0 is not the same as 'No significant differences were observed between the two evaluation methods'. Please rephrase this.

Authors' response: Following your request, this sentence has been amended for further clarity. Thanks:

Cohen's kappa test to assess the agreement among RECIST and irRECIST. A p-value below 0.05 indicates that the agreement among RECIST and irRECIST is statistically significant and indicates that the concordance between both is real and trustable.

Reviewer 3: The Clopper-Pearson method for CIs is not adequate for censored data.

Authors' response: This is a mistake while confecting the statistical section. Clopper-Pearson is used to calculate 95% CIs of ORR, which has no censored data. The statistical section has been amended for correctness:

The ORR were reported as percentage and their 95% CI was estimated using the Clopper–Pearson method. PFS and OS were estimated using the Kaplan–Meier method and Cox proportional model, and comparisons were tested using log-rank tests

Reviewer 3: Please specify the date of data cut-off for the analysis.

Authors' response: Data cut off was May 2024. Following your request data cut off is specified at first appearance in the manuscript:

Nine patients survived for at least four years from treatment initiation (Supplementary Figure 2), and eight were still alive at the time of data cutoff (May 2024).

Reviewer 3: Results on time-dependent analysis of imAEs (line 182/183): Please correct, it is the occurrence of imAEs and not the endpoint PFS/OS which is analysed as time-dependent factor.

Authors' response: The sentence has been modified according to your suggestions for further clarity:

Although patients with imAEs had numerically lower hazard ratios (HR) for both PFS and OS, these associations did not reach statistical significance when PFS and OS were analysed as time-dependent variables

Reviewer 3: Figure 3a/b waterfall plots: numbers do not add up, it should be 32 DTC patients etc but it is less.

Authors' response: For the waterfall plots, patients without target lesions, without measurable diameters or with best overall response equal to "UK" or "NE" were excluded. Excluding these patients, the numbers represented in Figure 3 are correct. Moreover, there were 30 patients with DTC, not 32.

Manuscript Number: NCOMMS-25-64273A

A phase II study of durvalumab plus tremelimumab for the treatment of patients with progressive, refractory advanced thyroid carcinoma - The GETNE-DUTHY trial.

AUTHORS' RESPONSE TO REVIEWERS

Reviewer #1 (Remarks to the Author)

This is a phase II, single-arm, multi-cohort trial was conducted determine whether dual anti-PD-L1/CTLA-4 inhibition using durvalumab and tremelimumab can improve the clinical outcomes in advanced thyroid cancer.

This is a much needed, invaluable study for advanced thyroid cancer and makes a tremendous contribution to what we have as treatment options for some of the aggressive thyroid cancers.

Thank you for clarifying the comments and suggestions. Given that a third of patients had received radiation before the drug combination, did you see any difference in survival of those who received radiation to neck before durval and treme combo versus who did not? Did any of these undergo surgery due to tumor shrinkage?

Authors' response: One-third of the patients with ATC (four patients) received radiotherapy prior to study inclusion. Of these, three did not undergo surgery, and one underwent surgery before receiving radiotherapy. Given the low number of patients with ATC (12 patients), the possibility to obtain significant conclusions from subgroup analysis were limited. Moreover, the two patients with longer survivorship did not receive previous radiotherapy, which could bias statistical tests interpretations. We identified in the Swimmer plot (Supplementary Figure 2) the patients who received prior radiotherapy, to give some information about the survival of patients with ATC who received and not previous radiotherapy.

Reviewer #1: Also one of the comments by author is that they didn't see any benefit of adding a second ICI over single agent. I dont think this is necessarily true as better results were seen from an OS standpoint with ipilimumab and nivolumab.

Authors' response: Thank you for your comments. Compared to the median OS in patients with DTC in our study, Sehgal et al. (2024) reported a higher median OS (24.6 vs. 13.8 months) with nivolumab plus ipilimumab therapy. However,

Oh et al. (2023) reported a median OS of 34.5 months in patients with DTC treated with pembrolizumab in monotherapy. The variability of the results obtained in the different trials reflects the underlying heterogeneity of the patients included in small thyroid trials, limiting the conclusions regarding the benefit of dual ICI versus monotherapy. We have modified the manuscript text accordingly.

Thus, with the data available, the benefit of adding anti-CTLA-4 to anti-PD-1 treatment cannot be confirmed or discarded~~does not have a clear benefit.~~

References:

Sehgal K, Pappa T, Shin KY, et al. Dual Immune Checkpoint Inhibition in Patients With Aggressive Thyroid Carcinoma: A Phase 2 Nonrandomized Clinical Trial. *JAMA Oncol.* 2024;10(12):1663-1671. doi:10.1001/jamaoncol.2024.4019

Oh DY, Algazi A, Capdevila J, et al. Efficacy and safety of pembrolizumab monotherapy in patients with advanced thyroid cancer in the phase 2 KEYNOTE-158 study. *Cancer.* 2023;129(8):1195-1204. doi:10.1002/cncr.34657

Reviewer #1: Overall, the paper looks ready for publication if they could add a comment of if there was any difference in survival in between those who received radiation to neck versus not.

Authors' response: Thank you for your revision. Regarding the survival of patients who received radiation and considering the low number of patients with ATC, we identified them in the Swimmer plot (Supplementary Figure 2), to give some information about the survival of patients with ATC who received and not previous radiotherapy.

Reviewer #3 (Remarks to the Author)

All my points have been addressed by the authors.

One minor issue: In the new suppl. tables 1 and 2, the table head should probably read 'estimated cumulative survival rate' instead of 'estimated cumulative survival ratio'

Authors' response: Thank you for your previous suggestions. They were helpful in improving the quality of our manuscript. Following your suggestion, we changed "ratio" by "rate" in Supplementary Tables 1 and 2.